# A genus in the bacterial phylum Aquificota appears to be endemic to Aotearoa-New Zealand

Jean F. Power [1], Carlo R. Carere [2], Holly E. Welford[3], Daniel T. Hudson[4], Kevin C. Lee [5], John W. Moreau [6], Thijs J. G. Ettema [7], Anna-Louise Reysenbach [8], Charles K. Lee [1], Daniel R. Colman [9], Eric S. Boyd [9], Xochitl C. Morgan[4,10], Ian R. McDonald[1], S. Craig Cary [1] ✉ & Matthew B. Stott [3] ✉

Allopatric speciation has been difficult to examine among microorganisms, with prior reports of endemism restricted to sub-genus level taxa. Previous microbial community analysis via 16S rRNA gene sequencing of 925 geothermal springs from the Taupō Volcanic Zone (TVZ), Aotearoa-New Zealand, revealed widespread distribution and abundance of a single bacterial genus across 686 of these ecosystems (pH 1.2-9.6 and 17.4-99.8 °C). Here, we present evidence to suggest that this genus, *Venenivibrio* (phylum Aquificota), is endemic to Aotearoa-New Zealand. A specific environmental niche that increases habitat isolation was identified, with maximal read abundance of *Venenivibrio* occurring at pH 4-6, 50-70 °C, and low oxidation-reduction potentials. This was further highlighted by genomic and culture-based analyses of the only characterised species for the genus, *Venenivibrio stagnispumantis* CP.B2[T], which confirmed a chemolithoautotrophic metabolism dependent on hydrogen oxidation. While similarity between *Venenivibrio* populations illustrated that dispersal is not limited across the TVZ, extensive amplicon, metagenomic, and phylogenomic analyses of global microbial communities from DNA sequence databases indicates *Venenivibrio* is geographically restricted to the Aotearoa-New Zealand archipelago. We conclude that geographic isolation, complemented by physicochemical constraints, has resulted in the establishment of an endemic bacterial genus.

A central tenet of biology is that geographic isolation reduces gene flow which can result in the evolution of new species (i.e., allopatric speciation)[1]. Historical models have proposed that the physical barrier of the ocean can limit migration to island ecosystems[2,3], provoking the divergence of distinct species that are geographically separated and initiating the formation of endemic taxa. While these models have been validated for macroorganisms across multiple biomes[4,5], evidence for microbial allopatry is largely constrained to strain-level

hot spring and hydrothermal vent lineages[6–10]. Microbial taxa in other non-extreme habitats are presented with fewer limits to dispersal[11–13], reinforcing the supposition that not all microorganisms diversify by the same mechanisms[14] and habitat fragmentation can acutely influence dispersal capability[15]. Gene flow, as a measure of genetic variability between conspecific taxa, can be interrupted by environmental and geographic isolation[14,16], with ecological niche boundaries contributing to speciation[17]. While there are some reports of endemism

occurring in microbial communities[18–20], evaluating diversification within populations has focused on strains[6,8,21], with no indication of genus-level endemism being observed[22].

The bacterial phylum Aquificota (formerly Aquificae)[23] was first reported from hot marine sediments in 1992[24], and is prevalent in geographically patchy, high temperature environments[25]. Growth temperatures for Aquificota isolates range from ~45 to 90 °C across both marine (i.e., deep-sea thermal vents) and terrestrial (i.e., geothermal springs) habitats, with most, but not all, strains dependent exclusively on hydrogen and/or sulfur species as reductant for metabolism[25]. Aquificota are also predominantly microaerophilic[25], further restricting the range of environmental conditions amenable to growth and successful dispersal between 'island-like' habitats. Indeed, ecological differentiation has even been suggested to induce sympatric speciation within an archaeal population of a single hot spring[26]. Thus, taxa from the Aquificota phylum are ideal candidates to investigate endemism, with ecosystems conducive to their growth facilitating speciation by allopatry, environmental selection and competition for available niches.

We recently published a comprehensive study on the microbial diversity and biogeography across 925 geothermal springs in the Taupō Volcanic Zone (TVZ), Aotearoa-New Zealand[27]. While pH was the principal driver of microbial diversity in geothermal spring communities spanning the entire 8000 km² region, Aquificota were the dominant taxa in springs >50 °C. Furthermore, three of the four most abundant genera in the study also belonged to Aquificota, with the genus *Venenivibrio* (family Hydrogenothermaceae) identified in 74.2% (*n* = 686) of geothermal springs analysed (Supplementary Data 1). Only one species of this genus, *Venenivibrio stagnispumantis* strain CP.B2ᵀ, has been formally described[28], with this strain isolated in 2007 from the Waiotapu geothermal field (*mana whenua*: Ngāti Tahu – Ngāti Whaoa), Aotearoa-New Zealand[29]. Interestingly, while sister genera of *Venenivibrio* (*Hydrogenothermus*, *Persephonella*, and *Sulfurihydrogenibium*) are globally distributed[9,30–36], we were unable to find any verifiable record of *Venenivibrio* taxa (cultivated species or via molecular signatures) outside of Aotearoa-New Zealand. The ubiquity of *Venenivibrio* within Aotearoa-New Zealand, but apparent absence of this taxon globally, led us to explore the hypothesis that the genus *Venenivibrio* is endemic to the Aotearoa-New Zealand archipelago.

To investigate the possibility of genus-level endemism, we undertook a detailed ecological study of Aquificota and *Venenivibrio* at a local scale (i.e., within Aotearoa-New Zealand) using both 16S rRNA gene amplicon and shotgun metagenomic data, as well as geothermal spring physicochemical measurements, while also searching for evidence of the *Venenivibrio* genus at a global scale (i.e., outside Aotearoa-New Zealand) in publicly available DNA sequence databases. Genome analysis identified distinguishing characteristics that may explain the exclusivity of this taxon to Aotearoa-New Zealand, with the type strain CP.B2ᵀ also cultured in spring waters in vitro from various locales to validate reported growth ranges and determine tolerances to physicochemical regimes of other geothermal provinces. Finally, phylogenomics confirmed placement of *Venenivibrio* as an independent genus, unequivocally distinguished from all other publicly available genomes and metagenome-assembled genomes (MAGs) within Hydrogenothermaceae, which was corroborated by the reconstruction of four *Venenivibrio* MAGs from TVZ springs. Our findings indicate *Venenivibrio* is only found in Aotearoa-New Zealand, with geography, associated dispersal limitation, and niche differentiation as possible mechanisms driving the apparent endemism of this bacterial genus.

## Results

### Aquificota and *Venenivibrio* ecology in Aotearoa-New Zealand
Of the 28,381 Operational Taxonomic Units (OTUs) generated from the 1000 Springs Project[27], 340 Aquificota-assigned OTUs were detected in 891 of the 925 individual geothermal springs analysed. Aquificota taxa comprised the Aquificaceae and Hydrogenothermaceae families (216 and 124 OTUs, respectively), with 111 *Venenivibrio*-assigned OTUs (Hydrogenothermaceae) found in 686 springs (Fig. 1a and Supplementary Data 1). The observed number of Aquificota OTUs increased with increasing spring temperature and pH (Fig. 1b). After conservatively removing exiguous Aquificota-assigned OTUs, 305 OTUs and their combined read abundance per 783 springs were then analysed to assess the effect of 46 physicochemical variables on diversity in the TVZ. Temperature, oxidation-reduction potential (ORP), pH, and conductivity had the greatest linear relationship to changes in both Aquificota OTU number and read abundance per spring (Fig. 2 and Supplementary Data 2). Multiple linear modelling suggested 38.9% of the variation in Aquificota OTU number was mostly attributable to temperature, ORP, and Ag (*p* < 0.001, Supplementary Data 3), with 35.8% of variation in read abundance predominantly related to temperature, ORP, pH, $SO_4^{2-}$, Mn, and $Fe^{2+}$ (*p* < 0.001; Supplementary Data 4). Temperature, pH, ORP, and $HCO_3^-$ most strongly correlated to Aquificota OTU number per spring, with comparable results produced for read abundance (*p* < 0.001, Pearson's and Spearman's correlation coefficients; Supplementary Data 5).

A reduced number of geothermal springs (*n* = 467), containing 99 *Venenivibrio*-assigned OTUs, were used to investigate the ecology of this genus in Aotearoa-New Zealand after again conservatively removing exiguous OTUs (Figure S1; Supplementary Data 6). Overall, the linear relationship between spring physicochemistry and the diversity of *Venenivibrio* appeared weaker than that of the entire phylum. Limited or no effect was observed for *Venenivibrio* OTU number and read abundance per spring with increased temperature and pH (Fig. 2 and Supplementary Data 7), with ORP and turbidity having the most association (Fig. 2 and Supplementary Data 7). Multiple linear modelling indicated Na, turbidity, and ORP principally accounted for 19.2% of variation in *Venenivibrio* OTU number across all springs (*p* ≤ 0.002; Supplementary Data 8), with 18.3% of variation in read abundance mostly influenced by ORP, turbidity, Li, and Na (*p* ≤ 0.001; Supplementary Data 9). Turbidity had the strongest correlation between both *Venenivibrio* diversity metrics, with ORP and $H_2S$ also having significant effects (*p* < 0.001, Pearson's and Spearman's correlation coefficients; Supplementary Data 10).

### Temperature, pH, and distribution of *Venenivibrio* in Aotearoa-New Zealand
Polynomial regression highlighted *Venenivibrio* read abundance was greatest within a limited range of pH and temperature conditions (Fig. 3a). From the reduced dataset of 467 springs, microbial communities which contained ≥45% of *Venenivibrio*-assigned reads (*n* = 86) had a median pH and temperature of 6.0 (IQR 1.3) and 64.8 °C (IQR 15.2), respectively (Fig. 3a). In comparison, springs with <45% of *Venenivibrio* reads (*n* = 381) had a median pH and temperature of 6.1 (IQR 4.3) and 61.9 °C (IQR 41.3), respectively. The narrow pH and temperature range for *Venenivibrio* abundant springs was again evident when springs were subdivided into environmental increments, with the greatest read abundance observed at pH 4–6 (Fig. 3b) and 50–70 °C (Fig. 3c). The 10 most abundant *Venenivibrio* OTUs were also found in more acidic and hotter spring environments (≥10% relative abundance per spring community; pH 5.8 and 67.2 °C; Fig. 4) than their less abundant counterparts (<1000 reads; pH 6.0 and 62.9 °C; Fig. S2).

Considering the distribution of *Venenivibrio* across the TVZ, average relative read abundance in springs containing the taxon per geothermal field ranged from 0.8 to 35.7%, with Waiotapu having the greatest abundance (Fig. 5a). Twenty individual springs located in six geothermal fields were found to have ≥85% of the total microbial community assigned to *Venenivibrio* (Fig. S3), with a median pH and temperature of 5.5 and 66.0 °C, respectively. In contrast to a weak distance-decay pattern observed in microbial communities across the TVZ[27], *Venenivibrio* populations did not show this trend (*n* = 99 OTUs, $R^2$ = 0, *p* = 0.007; Fig. S4), with 19 of the most abundant OTUs found in

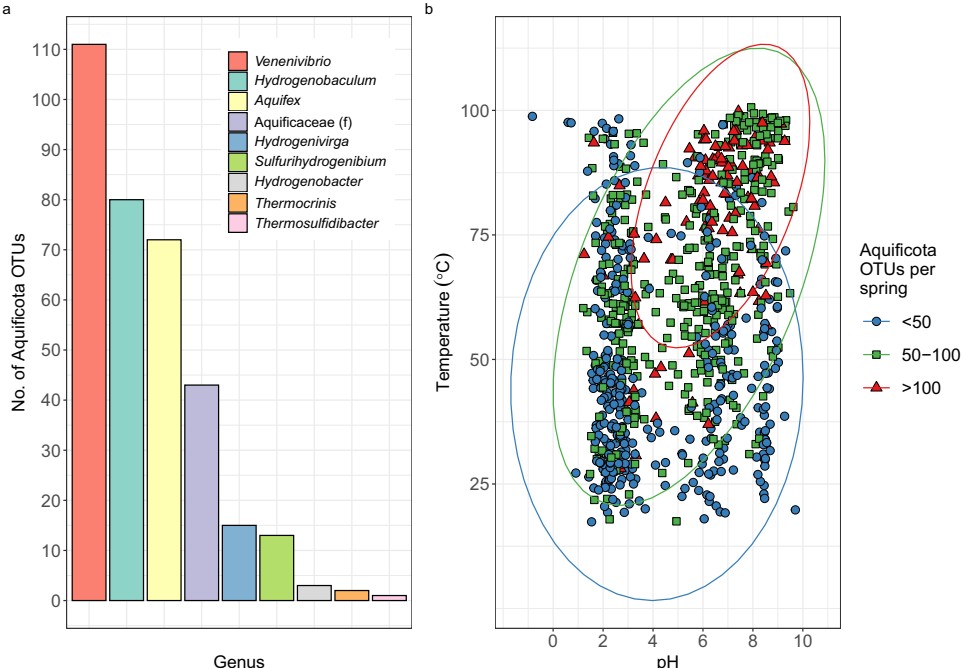

**Fig. 1 | Diversity of Aquificota 16S rRNA genes in the Taupō Volcanic Zone (TVZ), Aotearoa-New Zealand. a** The division of Aquificota-assigned 16S rRNA gene operational taxonomic units (OTUs) to their respective genus (defined by colour) from amplicon sequencing of the microbial communities found in 925 geothermal springs prior to filtering. Where taxonomy was unable to be assigned to an OTU at genus level, the corresponding family is shown (f). **b** All springs that contained Aquificota (n = 891) in the TVZ are plotted according to environmental pH and temperature. Diversity in each spring is represented by the number of Aquificota-assigned OTUs defined by colour and shape (blue circles [<50 OTUs], green squares [50-100 OTUs], or red triangles [>100 OTUs]), with data ellipses assuming multivariate t-distribution and a 95% confidence level.

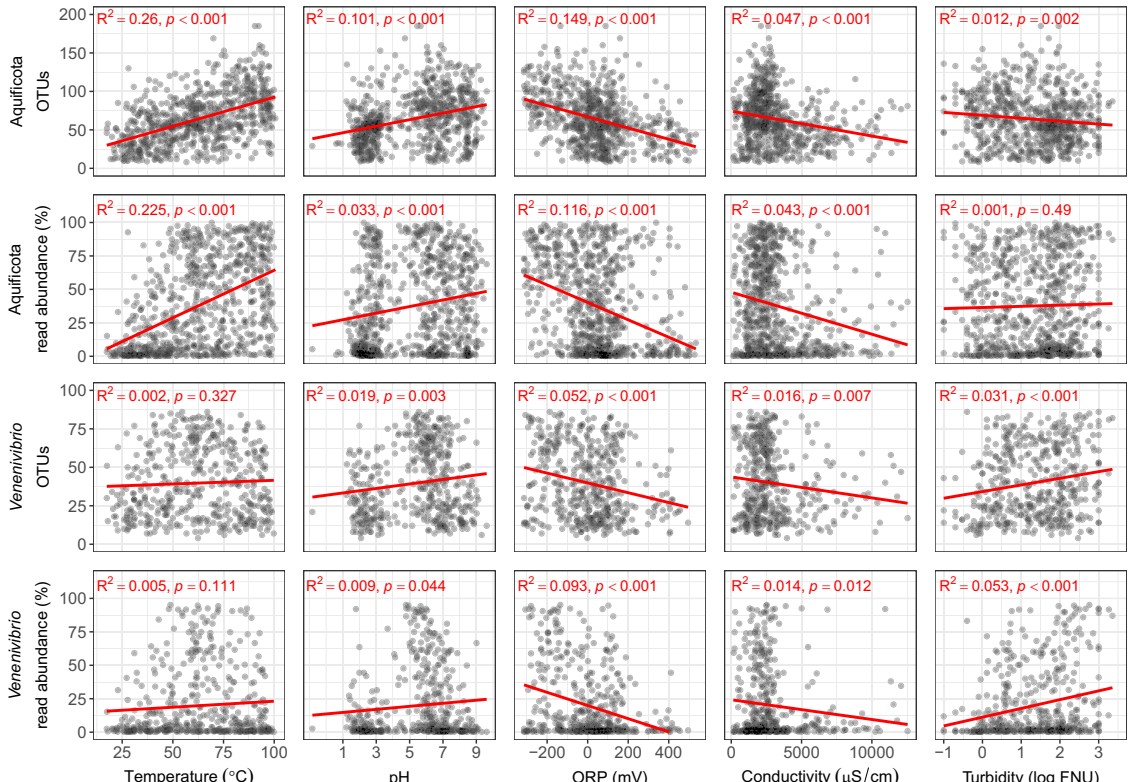

**Fig. 2 | Aquificota and *Venenivibrio* 16S rRNA gene diversity as a function of select environmental measurements in Aotearoa-New Zealand geothermal springs.** Diversity is represented as the number of operational taxonomic units (OTUs) assigned to either the phylum or genus in each spring, and the relative read abundance of Aquificota- or *Venenivibrio*-assigned reads per spring community (%). Linear regression was applied to each diversity metric against spring temperature, pH, oxidation-reduction potential (ORP), conductivity, and turbidity. *Venenivibrio* OTU number and read abundance strongly correlated with each other (Pearson's and Spearman's coefficients: 0.674 and 0.714, respectively, p = 0.000; Supplementary Data 10).

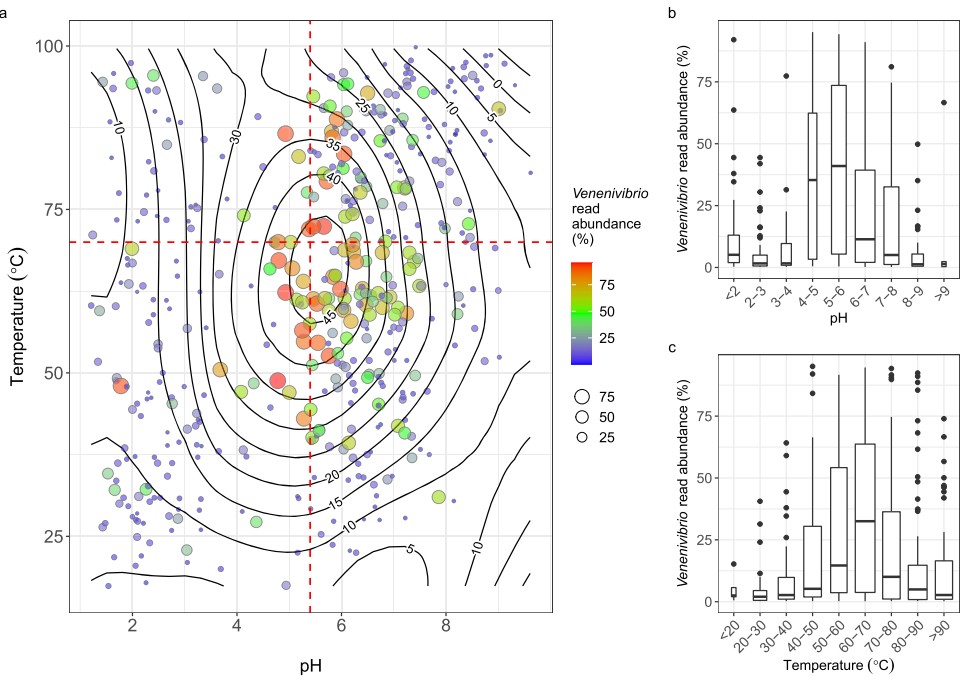

**Fig. 3 | Relative read abundance of *Venenivibrio* 16S rRNA genes as a function of geothermal spring pH and temperature in Aotearoa-New Zealand. a** A local polynomial regression of *Venenivibrio* 16S rRNA gene relative read abundance (contours) was applied to examine the relationship to both spring pH and temperature (*n* = 466), with relative abundance also described by colour and size of the points. The dashed red lines represent the reported optimum pH and temperature of the type strain, *V. stagnispumantis* CP.B2[T], as per Hetzer et al.[28] **b** The relative read abundance of *Venenivibrio* 16S rRNA genes in geothermal springs (post filtering; *n* = 467) split by increments of spring pH. The centre line of the boxplots

represents the median, the box edges correspond to the first and third quartiles, and the whiskers extend to 1.5 times the interquartile range from the edges. Outliers beyond these values are presented as individual points. **c** The relative read abundance of *Venenivibrio* 16S rRNA genes in geothermal springs (post filtering; *n* = 467) split by increments of spring temperature. The centre line of the boxplots represents the median, the box edges correspond to the first and third quartiles, and the whiskers extend to 1.5 times the interquartile range from the edges. Outliers beyond these values are presented as individual points.

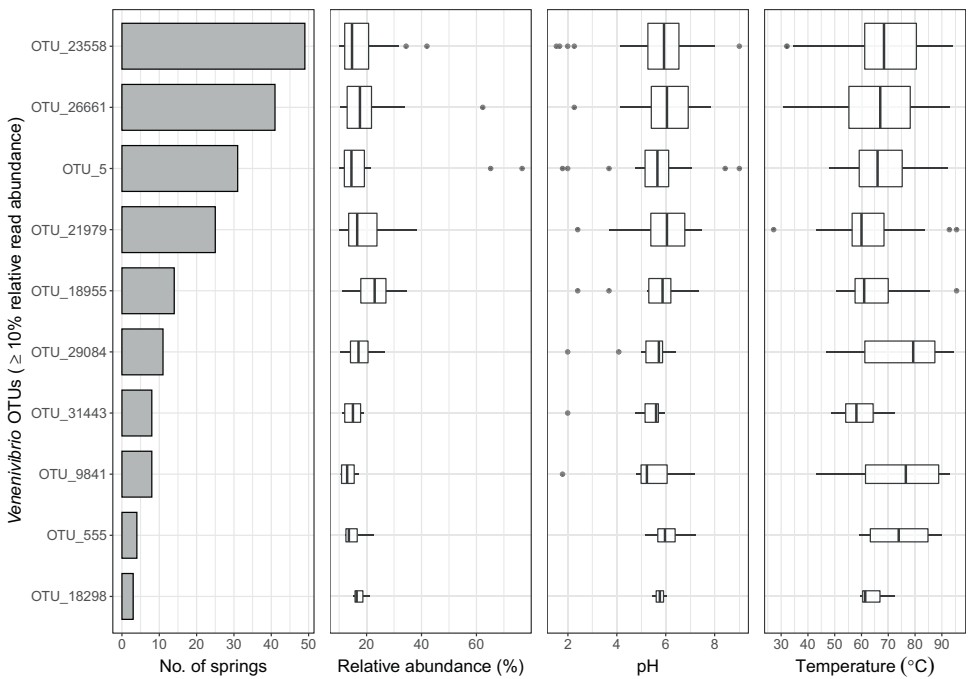

**Fig. 4 | Prevalence, read abundance, pH, and temperature of putative *Venenivibrio* microdiversity via operational taxonomic units (OTUs).** Only OTUs with relative read abundances of ≥10% per spring community (*n* = 10 OTUs) that assigned to the genus *Venenivibrio* are shown. Abundant OTUs are ordered via prevalence (i.e., no. of springs) across all springs where *Venenivibrio* was found

(post-filtering; *n* = 467 geothermal springs). Median pH and temperature for abundant OTUs were 5.8 (IQR 0.9) and 67.2 °C (IQR 17.4), respectively. The centre line of the boxplots represents the median, the box edges correspond to the first and third quartiles, and the whiskers extend to 1.5 times the interquartile range from the edges. Outliers beyond these values are presented as individual points.

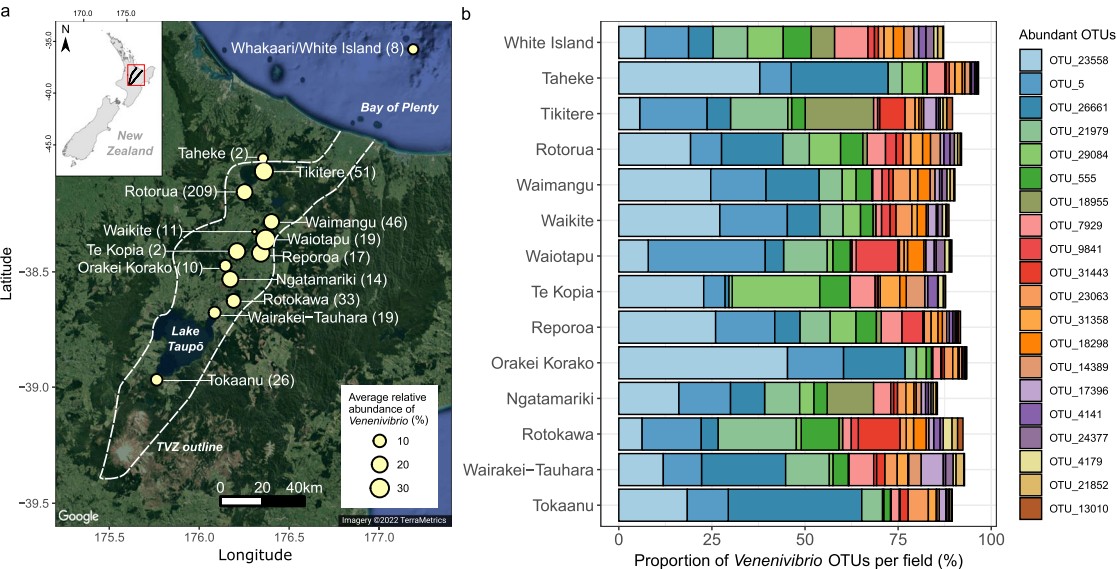

**Fig. 5 | Distribution of *Venenivibrio* 16S rRNA genes in Aotearoa-New Zealand.**
**a** A map of the Taupō Volcanic Zone (TVZ), with geothermal fields where *Venenivibrio* 16S rRNA genes were detected highlighted in yellow. Geothermal fields are scaled in size according to average relative abundance of *Venenivibrio*-assigned sequencing reads across spring communities in each field, with the number of springs containing *Venenivibrio* displayed in brackets. **b** The proportion of the 20 most abundant *Venenivibrio* 16S rRNA gene operational taxonomic units (OTUs) per geothermal field are displayed in colour, where 100% equals the total relative abundance of *Venenivibrio* reads within each field. Geothermal fields are ordered north to south. A breakdown of this relative abundance in reads per spring can be found in Fig. S5. Map data ©2022 Google.

all 14 geothermal fields analysed (Fig. 5b). In particular, Whakaari/White Island, situated ~48 km offshore from the mainland, showed an increased even spread *Venenivibrio* OTUs in spring communities when compared to other fields. Conversely, most of the *Venenivibrio* read abundance in two springs from the Waiotapu geothermal field was assigned to a single OTU (OTU_5; 65 and 77% of the total microbial community; Figs. 4 and S5). While OTU_5 was the most prevalent *Venenivibrio*-OTU in springs across the entire TVZ (*n* = 456; Supplementary Data 6), its next greatest read abundance in an individual spring was 22% of the microbial community. The third greatest read abundance of a single *Venenivibrio* OTU was OTU_26661, which comprised 62% of a Rotorua spring community (Figs. 4 and S5). All remaining OTUs had <42% relative read abundance in any of the 467 microbial communities analysed.

### Genome annotation of *V. stagnispumantis* CP.B2[T]
The CP.B2[T] genome encoded the capacity for a chemolithoautotrophic mode of energy generation and carbon fixation (Supplementary Data 11). This included genes for the Type I reductive tricarboxylic acid (rTCA) cycle[37], cytochrome bd (*cydAB*) as the sole respiratory terminal oxidase, and two [NiFe]-hydrogenases from groups 1b (*hynAB*) and 2d (*huaSL*)[38]. Despite obligately requiring a reduced source of sulfur for growth[28], predicted sulfur-metabolising genes were limited to two copies of a membrane-bound sulfide:quinone oxidoreductase (*sqr*). There was evidence of an extensive sulfur-trafficking network inside the cell, including three copies of a TusA-related sulfurtransferase, two rhodanese-type transferases (*sseA* and *pspE*), and the *dsrEFH* complex[39,40]. There was no evidence of capacity for either assimilatory sulfate reduction or the full SOX pathway for sulfur/thiosulfate oxidation. Genes for arsenic resistance (*arsRBC*) were detected, which presumably transports reduced arsenite [As$^{3+}$] or stibnite [Sb$^{3+}$] out of the cytosol via a membrane-bound efflux pump[41,42]; both As$^{3+}$ and Sb$^{3+}$ are present in elevated concentration in the CP.B2[T]-host environment[43]. Consistent with the characterisation of CP.B2[T] and an investigation into microbial-induced arsenic cycling of Champagne Pool[28,44], no evidence of described genes encoding dissimilatory arsenic metabolism was noted in the genome. Annotation also

confirmed flagellar assembly and a range of chemotaxis genes (Supplementary Data 11).

Analysis of ten Hydrogenothermaceae genomes (including the *Venenivibrio* type strain, *V. stagnispumantis* CP.B2[T]) to identify any distinguishing characteristics contributing to the apparent endemism of *Venenivibrio* revealed all genomes encoded ATP-citrate lyase (ACL; gene *aclAB*), used in the reductive TCA cycle to fix carbon dioxide. Interestingly, citrate synthase genes (*gltA*), required in the oxidative TCA cycle, were exclusive to *P. marina* EX-H1[T], *P. hydrogeniphila* 29W[T], and all *Sulfurihydrogenibium* spp. analysed. Most notably, CP.B2[T] was the only isolate that did not encode the full SOX pathway. Collectively, these differences suggest a decreased metabolic capability for CP.B2[T] compared to other Hydrogenothermaceae isolates. Additional interpretation of genome annotation is detailed in Supplementary Note 1.

### Growth range reassessment of *V. stagnispumantis* CP.B2[T]
In light of the broad distribution of *Venenivibrio* across a range of temperature and pH conditions, we reinterrogated in vitro growth conditions of the type strain *V. stagnispumantis* CP.B2[T]. Results from these experiments differed considerably from those previously reported (Supplementary Data 12)[28]. Notably, viable growth was observed (via floc formation) up to 8% w/v NaCl (previously reported as 0.8% w/v NaCl across a pH range of 3.0–8.5 (previously reported as pH 4.8–5.8). However, reassessed growth optima for temperature (70.4 °C), pH (pH 6.0), salinity (0–0.2% w/v), and O$_2$ (<1.25–10% v/v) tolerance did not change substantially (Supplementary Data 12), corresponding with the preferred conditions for increased abundance of the taxon as demonstrated in Fig. 3. Further, to test whether there were some cryptic physicochemical conditions that restricted *Venenivibrio* growth outside Aotearoa-New Zealand, we tested growth in filtered spring water from Obsidian Pool, Yellowstone National Park (YNP), USA, and Champagne Pool (TVZ) as a control. Obsidian Pool displays similar physicochemistry to Champagne Pool and is known to support resident populations of *Sulfurihydrogenibum*[32]. CP.B2[T] grew in all conditions (natal spring water, spring water with supplemented S$^0$, and spring water with supplemented S$^0$, NH$_4$Cl, and

$KH_2PO_4$) for both geothermal features. This section is expanded in Supplementary Note 2.

## Global search for 16S rRNA genes reported as, or closely related to, *Venenivibrio*

No significant evidence was found of the 16S rRNA gene of *V. stagnispumantis* CP.B2[T] outside of Aotearoa-New Zealand (Fig. S6 and Supplementary Data 13–15). Over 971,117 samples, 26.7 billion 16S rRNA gene sequences, and 12.2 petabytes of sequence data were exhaustively searched from the following nine microbial databases: NCBI's Nucleotide collection[45], the Sequence Read Archive (SRA)[45], SILVA[46], the Ribosomal Database Project (RDP)[47], Greengenes[48], the Integrated Microbial Next Generation Sequencing (IMNGS) platform[49], JGI's Integrated Microbial Genomes and Microbiomes (IMG/M) system[50], the Earth Microbiome Project (EMP)[51], and the Qiita platform[52]. Putatively identified *Venenivibrio* 16S rRNA gene sequences were invalidated via manual interrogation of sequence similarity to *V. stagnispumantis* CP.B2[T] (<95%) and other Hydrogenothermaceae isolates. The literature search for the word '*Venenivibrio*' in all NCBI databases (including PubMed) and Google Scholar also yielded no significant presence of the genus outside the Aotearoa-New Zealand archipelago. Additionally, maximum-likelihood phylogenetic analysis of the SILVA SSU r138.1 database entries originally classified as *Venenivibrio* clustered only Aotearoa-New Zealand-based sequences within the genus (Fig. S6). Detailed results on each database search are outlined in Supplementary Note 3.

## Screening metagenomes for *Venenivibrio*

All 16 metagenomic samples from TVZ geothermal springs (i.e., local metagenomes) contained genomic signatures of *V. stagnispumantis* using the taxonomic classification tool Kraken2[53], with a maximum read abundance of 45.1% *Venenivibrio* per community (Supplementary Data 16). Champagne Pool metagenomes ($n = 4$), the locale where CP.B2[T] was originally isolated, had read abundances between 8.6 to 27.7% across the outflow channel, adjacent terrace, rim of the spring, and water column. A total of 188 metagenomes were associated with the term 'hot spring' (i.e., global metagenomes) from a database of 20,206 terrestrial samples[54]. The terms 'hotspring' and 'geothermal' yielded no results, while 'hydrothermal' resulted in 16 non-surface samples (i.e., deep-sea vent or subsurface habitats) that were excluded from classification analysis. Twenty-one of the 188 hot spring metagenomes returned putative traces of *Venenivibrio* from community taxonomic analysis using Kraken2[53] (Supplementary Data 17). Eight of these exhibited 0.01-1.08% of the total community being assigned to *V. stagnispumantis*, with the remaining 13 metagenomes having negligible *Venenivibrio* read abundance (0%) compared to each respective community.

To test the validity of these putative traces from the Kraken2 classification, four local and six global metagenomes with *Venenivibrio*-assigned reads were aligned to both the *V. stagnispumantis* CP.B2[T] genome and the most closely related Hydrogenothermaceae (Fig. 6a). All metagenomes from Aotearoa-New Zealand mapped to 91.8–96.8% of the CP.B2[T] genome, with an average coverage depth of 35-1006× (Fig. 6a and Supplementary Data 18). In contrast, the three other Hydrogenothermaceae genomes had relatively minimal average coverage breadth (≤0.81%) and depth (≤0.17×) by the local metagenomes (Fig. 6a and Supplementary Data 18). The global metagenomes, sourced from geothermal springs in Japan ($n = 3$), USA ($n = 2$) and Canada ($n = 1$), aligned to 0.1–10.2% of the CP.B2[T] genome, with an average coverage depth of 0.0–16.2× (Fig. 6a and Supplementary Data 18), leading us to conclude *Venenivibrio*, or any divergent strain within this genus, were not present in these global metagenomes. These samples had slightly more coverage breadth across the other Hydrogenothermaceae genomes, mainly *Sulfurihydrogenibium* sp. Y03AOP1 (0.2–13.1%) and *S. yellowstonense* SS-5[T] (0.4–12.9%). Three of

the global metagenomes (sample IDs DRR163686, DRR163687, and SRR7905022) did cover 85-98% of a single *Venenivibrio* contig (IMG Scaffold ID 2724812627 and Locus Tag Ga0170441_134; Fig. 6a), which encodes 5S, 16S, and 23S rRNA genes, along with tRNAs for alanine and isoleucine.

Additionally, metagenome-assembled genomes (MAGs; Supplementary Data 19) were created from four local and two global metagenomes with the greatest number of reads putatively assigned (via Kraken2; Supplementary Data 16 and 17) and mapped (via bowtie2; Supplementary Data 18) to *Venenivibrio*. Each of the local metagenomes produced a MAG that had an average nucleotide identity (ANI) of 94.8–97.9% to the *V. stagnispumantis* CP.B2[T] genome. Conversely, MAGs from the two global samples with the closest congruity to CP.B2[T] were ANI < 80% similar. Completeness for all MAGs ranged from 91.9–99.2% (Supplementary Data 19). Mock communities designed to test the robustness and credibility of metagenomic analysis substantiated the classification and alignment of *Venenivibrio* in these metagenomic samples (Fig. S7 and Supplementary Data 20 and 21), with expanded details on this in Supplementary Note 4.

## Phylogenomics of Hydrogenothermaceae

Thirty-eight publicly available genomes and MAGs ($n = 11$ and 27 respectively) from the Hydrogenothermaceae were used to build an approximate maximum-likelihood phylogenomic tree, placing *V. stagnispumantis* strain CP.B2[T] as a separate clade to all other genera (Fig. 6b). GTDB also positioned CP.B2[T] as a novel genus within the Hydrogenothermaceae (RED value of 0.82). The four MAGs from Aotearoa-New Zealand geothermal springs positioned within this clade, with the two global MAGs generated by this study phylogenomically classifying as *Sulfurihydrogenibium* spp. Pairwise ANIs of Hydrogenothermaceae isolates to CP.B2[T] were <80%, with *Sulfurihydrogenibium yellowstonense* SS-5[T] being the mostly closely related non-Aotearoa-New Zealand taxon. Molecular clock analysis between CP.B2[T] and SS-5[T] suggested divergence of the most recent common ancestor to the two genera occurred ~67 mya.

## Discussion

### *Venenivibrio* is ubiquitously dispersed in Aotearoa-New Zealand, but not globally

The phylum Aquificota dominates geothermal spring water in the TVZ, a region spanning 8,000 km², with *Venenivibrio* identified as the most abundant genus across the 925 springs analysed[27]. Illustrating that local dispersal within the TVZ is not limited, *Venenivibrio* was found in 74.2% of springs across a wide range of physicochemical conditions. While the reassessed growth ranges for the type strain, *V. stagnispumantis* CP.B2[T], expand the environmental conditions under which we could conceivably expect growth or persistence, this does not necessarily account for the physicochemical ranges where we detect *Venenivibrio* (pH 1.2-9.6, 17.4-99.8 °C; Fig. 3a). Amplicon sequence detection does not infer cell viability in springs where *Venenivibrio* densities are low[55], however, increasing the pool of cultivated *Venenivibrio* isolates will expand understanding of the dispersal potential and range of phenotypes possessed by the taxon. The presence of 111 *Venenivibrio* OTUs also suggests that there is a high degree of microdiversity within this genus[56], with diversification most likely initiated by environmental partitioning. Nonetheless, the lack of a distance-decay pattern for *Venenivibrio* populations within the TVZ, and the distribution of *Venenivibrio*-assigned OTUs across all geothermal fields in the region, reinforce unlimited dispersal on a local scale and substantiates the pervasiveness of the genus within Aotearoa-New Zealand.

Twenty 'hotspots' of increased *Venenivibrio* read abundance were identified in the TVZ that may serve as reservoirs to facilitate local dispersal, a mechanism recently hypothesised to support colonisation of newly-formed volcanic habitats in the Kingdom of Tonga[57].

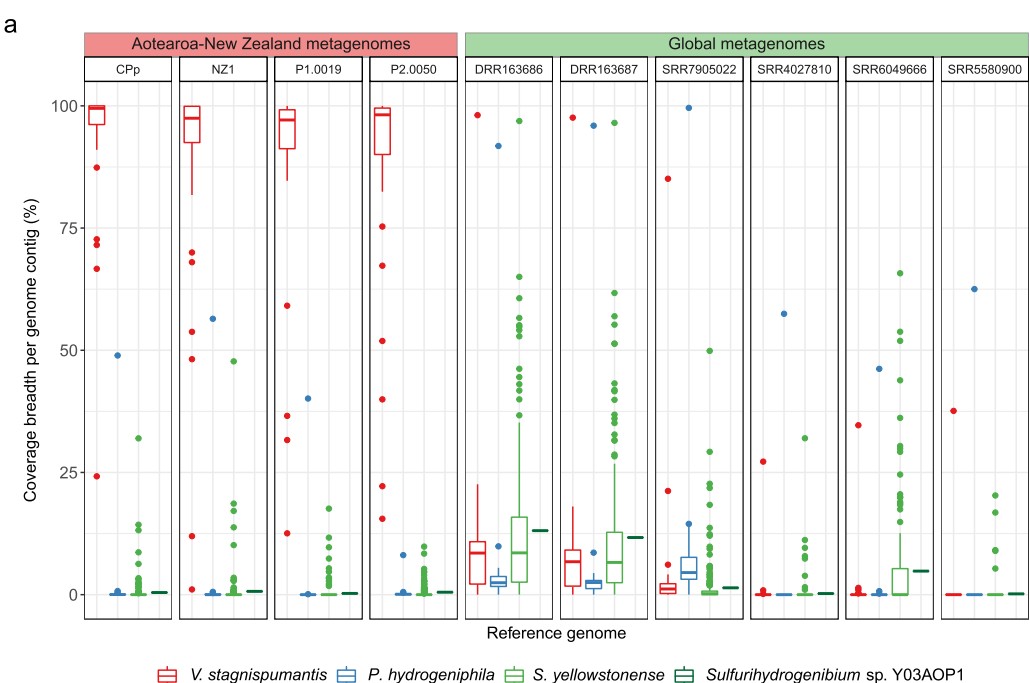

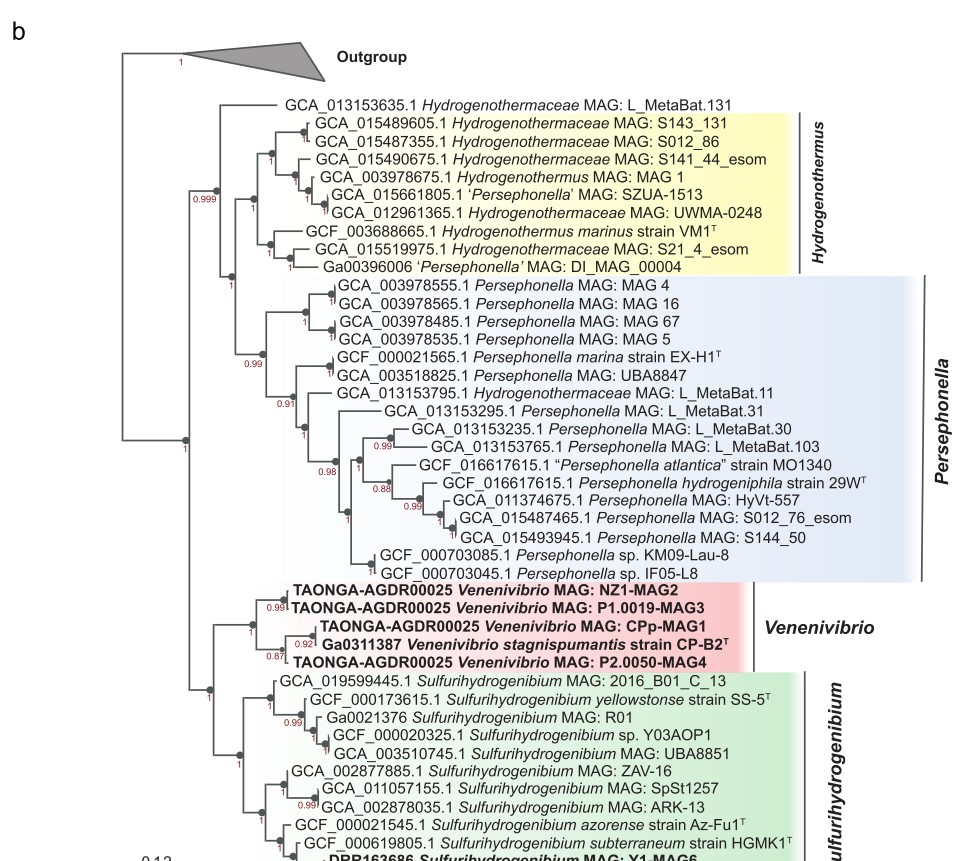

The planktonic lifestyle of *Venenivibrio*, corroborated by our exclusive use of water column samples[27], the source material for the isolation of CP.B2[T][29], and annotation of a full suite of flagellar and chemotaxis genes in the genome, likely contributes to this local dispersal via subsurface geothermal aquifers and/or water vapour, enhanced by increased read abundance of the taxon in springs hot enough to generate steam (50–70 °C). Interestingly, *Venenivibrio* was not reported as an abundant community member of geothermal spring sediments and sinters (*n* = 138) from the TVZ[58], which supports the planktonic lifestyle model promoting dissemination at a local scale.

An intensive search for *Venenivibrio* in both amplicon and metagenomic datasets did not yield any noteworthy results from outside Aotearoa-New Zealand. While a small number of global metagenomes initially classified trace levels of sequences as *Venenivibrio*, comprehensive interrogation refuted their classification within this genus. Alignment of global metagenomes to Hydrogenothermaceae genomes

**Fig. 6 | Metagenomic evidence for *Venenivibrio* in Aotearoa-New Zealand.**
**a** Metagenome reads derived from four Aotearoa-New Zealand (left) and six global (right) geothermal spring microbial communities were mapped to the *V. stagnis-pumantis* CP.B2$^T$ genome and three genomes of the most closely related strains from family Hydrogenothermaceae. Box plots represent mapped coverage breadth across each of the reference genome contigs in colour (*V. stagnispumantis* CP.B2$^T$ [red, n = 43 contigs], *P. hydrogeniphila* 29W$^T$ [blue, n = 19 contigs], *S. yellowstonense* SS-5$^T$ [light green, n = 228 contigs], and *Sulfurihydrogenibium* sp. Y03AOP1 [dark green: n = 1 contig]) by the 10 metagenomic samples. The centre line of the box-plots represents the median, the box edges correspond to the first and third quartiles, and the whiskers extend to 1.5 times the interquartile range from the edges. Outliers beyond these values are presented as individual points. Samples are ordered by decreasing sequence similarity to CP.B2$^T$. Average coverage breadth and depth across full genomes are presented in Supplementary Data 18. **b** An approximate maximum-likelihood phylogenomic tree of all publicly available genome assemblies, including isolates and metagenome-assembled genomes (MAGs), from Hydrogenothermaceae. Representative genera of the family are emphasised by colour; *Venenivibrio* (red), *Persephonella* (blue), *Sulfurihy-drogenibium* (green), and *Hydrogenothermus* (yellow). Four Aotearoa-New Zealand and two global MAGs generated in this study are also highlighted in bold. The tree was built using 49 house-keeping COG gene families, with bootstrap confidence values indicated at nodes. Selected members of Aquificaceae (*Aquifex aeolicus* VF5$^T$ [GCF_000008625.1], *Hydrogenobacter thermophilus* TK-6$^T$ [GCF_000010785.1], and *Thermocrinis ruber* OC 1/4$^T$ [GCF_000512735.1]) were included as an outgroup.

corroborated their lack of *Venenivibrio*, with MAGs created from these samples having only ≤80% similarity to *V. stagnispumantis* CP.B2$^T$. In general, reduced microbial biomass in geothermal spring water columns, compared to associated sediments[59], has fostered a sampling bias of these ecosystems. While this bias could partially explain the absence of reported *Venenivibrio* in datasets outside Aotearoa-New Zealand, dominant planktonic taxa are usually detectable (in lower abundance) within sediments from the same geothermal feature[59–61]. Therefore, despite prior evidence supporting the aeolian transport of some cosmopolitan taxa to reach geographically isolated habitats over geological time[18,20], our findings show *Venenivibrio* has not adhered to this process of community assembly.

We propose that dispersal of *Venenivibrio* outside of Aotearoa-New Zealand is limited. Local-global overlap in microbial diversity depends on multiple factors including: community size, global diversity, inter-patch environmental heterogeneity, and patch connectivity[62]. Aotearoa-New Zealand exists as an isolated archipelago within the South Pacific Ocean; near the convergent boundary of the Pacific-Australian tectonic plates. The nearest substantial land-masses are New Caledonia to the northwest (~1400 km) and Australia to the west (~1900 km). Contrary to the comparatively small TVZ, Aotearoa-New Zealand's isolated position within the Pacific Ocean restricts *Venenivibrio* from quickly reaching optimal habitats off-shore via atmospheric transport, allowing environmental filtering to limit dispersal globally. Although there is ample evidence of global microbial aeolian dispersal[18,63], strong selection occurs during long-range atmospheric transport[64], with stress tolerance being an important trait to aid disseminsation[13]. Neither *Venenivibrio* nor any known members of Aquificota produce spores or cysts that would increase stress tolerance and survival over prolonged periods of travel. Other constraining factors include the prevailing westerly winds in Aotearoa-New Zealand that would drive aerosol dispersal east, where the nearest substantial land mass is the South American continent. Ocean salinity is likely to further restrict *Venenivibrio* distribution, with optimal growth of the type strain CP.B2$^T$ observed at 0.0–0.2% w/v NaCl. Sodium prominently featured in modelling of all 46 environmental parameters from geothermal springs that contained *Venenivibrio*, indicating diversity was affected by concentrations of this solute. Additionally, no traces of *Venenivibrio* (or indeed any Aquificota) were found in a global aero-sol dataset of 596 near ground air, soil, and high-altitude microbial samples[65], indicating that substantial atmospheric transport is likely not conducive to this phylum. Intriguingly, growth of CP.B2$^T$ in Obsidian Pool spring water (YNP) indicates that if viable cells do reach geothermal spring ecosystems outside Aotearoa-New Zealand, contemporary physicochemical conditions could allow *Venenivibrio* establishment, so long as the environmental niche does not considerably overlap with those of resident organisms. Nevertheless, the inability for the taxon to spread is exacerbated by a lack of suitable niches proximal to the TVZ. This prohibits the growth of sufficient cell numbers to create the 'stepping-stones' required to facilitate migration globally.

## Specific growth requirements facilitate *Venenivibrio* to be locally abundant, but globally stranded

Maximal diversity of *Venenivibrio* occurs in reducing, hypoxic geo-thermal springs, with elevated concentrations of arsenic, antimony, and hydrogen sulfide, and with a limited pH and temperature range. The apparent specificity of *Venenivibrio*'s preferred growth environ-ment was confirmed by the identification of ORP and turbidity as the main parameters associated with increased read abundance of the taxon and reinforces dispersal limitation beyond geothermal fields through strict selection of appropriate environmental niches. The association with turbidity is slightly confounding as the most parsi-monious explanation is that *Venenivibrio* associates with particles, an observation not supported by a recent study of TVZ sediments[58]. However, widespread distribution of *Venenivibrio* in the TVZ was also observed in springs far outside previously reported physicochemical optima[28] and was corroborated by our growth range re-evaluation of the type strain, *V. stagnispumantis* CP.B2$^T$. Either initial characterisa-tion did not capture the full growth capabilities of the type strain, some type of cross-feeding or dependency is occurring with other commu-nity members, and/or there is much more microdiversity and physi-cochemical preferences at species/strain level represented by the 111 *Venenivibrio*-assigned OTUs found in our analysis[66]. Nevertheless, springs abundant with *Venenivibrio*-assigned reads still confirmed a preference for limited physicochemical regimes, in particular those with the infrequently encountered pH 4–6[67], further supporting that the genus requires a specialised habitat to thrive.

These specific growth characteristics were substantiated by gen-ome annotation of *V. stagnispumantis* CP.B2$^T$, which demonstrated a streamlined metabolism characterised by both autotrophy and microaerophily. The obligatory reliance on hydrogen as an electron donor (via Type 1b and 2d NiFe-hydrogenases) suggests a lack of metabolic versatility to utilise other energy sources that are often present in hot springs, with no fully characterised pathway for sulfur oxidation found. Comparative genomics within the Hydro-genothermaceae validated this metabolically constrained lifestyle, with *Sulfurihydrogenibium* spp., in particular, encoding three enzymes/ pathways (citrate synthase as part of the oxidative TCA cycle, a full SOX system for thiosulfate/sulfur oxidation, and a cytochrome-cbb3 terminal oxidase) conspicuously absent from the CP.B2$^T$ genome. Contrasting the prominence of planktonic *Venenivibrio* in TVZ spring water columns over sediments[27,58], *Sulfurihydrogenibium* spp. also seem to dominate within filamentous mat communities in geothermal areas[33,36,68–70]. This dichotomy in growth media between the genera, accompanied by the additional metabolic flexibility found in *Sulfur-ihydrogenibium* spp., could have formed alternate environmental niches, a mechanism that can aid speciation[17]. The reduced number of hydrogenases in *Sulfurihydrogenibium* spp. isolated from YNP[32], for example, highlights that these taxa appear to have dissimilar metabolic strategies. These differing traits suggest environmental selection and niche differentiation may have induced ancestral divergence, followed by speciation, within the Hydrogenothermaceae family, which was either driven or reinforced by geographic isolation of these habitats.

### Is *Venenivibrio* an endemic genus to Aotearoa-New Zealand?

Identification of *Venenivibrio* in 686 amplicon and 16 metagenomic samples from Aotearoa-New Zealand geothermal springs categorically confirmed the presence of the genus in the country; with four of the derived MAGs having >95% similarity to *V. stagnispumantis* CP.B2[T]. The type strain, CP.B2[T], was isolated from Champagne Pool, Waiotapu, in the TVZ[29] and no other strain from either genus or species has been validly described. The first evidence of this genus was a clone reported in 2003 from a geothermal spring in Kuirau Park, Aotearoa-New Zealand[71] and since then, all 16S rRNA gene sequences identified with ≥95% nucleotide similarity have originated from Aotearoa-New Zealand[27,44,72]. A phylogenomic analysis of Aquificota genomes further confirmed CP.B2[T] as a distinct genus within the Hydrogenothermaceae, with only Aotearoa-New Zealand-generated MAGs included within this clade. All other global genomes and MAGs from the family placed within either the *Sulfurihydrogenibium*, *Persephonella*, or *Hydrogenothermus* genera, strengthening the domestic exclusivity of *Venenivibrio* to Aotearoa-New Zealand.

A recent study analysing 36,795 bacterial and archaeal genomes from ~7000 locations around the world found most species (and even strains; average nucleotide identity ≥99.9%) are globally distributed, but suggested continental-scale endemism does occur at a sub-strain taxonomic level[22]. Hot spring and subsurface lineages, however, displayed the slowest rates of dissemination in this analysis[22], reinforcing the concept that these ecosystems can act as 'isolated islands' of microbial evolution[73]. Endemism has long been suggested for thermophiles in geothermal springs, where physical constraints and extreme physicochemistry create discrete, isolated microbial islands[6,7]. Indeed, endemism has often been reported as a function of genetic resolution[74] but there has yet been a description of genus-level endemism of microorganisms in either extreme or non-extreme environments. Given the specific growth requirements of *Venenivibrio*, multiple dispersal limitations, and the geographical setting of Aotearoa-New Zealand, it is plausible for endemism to occur within this genus. It is also important to highlight that time has been proposed as a main influence on building microbial community diversity in any ecosystem[22,75]. Indeed, a molecular clock analysis suggests *Venenivibrio* diverged from its sister genus *Sulfurihydrogenibium* ~67 mya, approximating a similar timeline to when the Zealandia landmass split from Gondwana via seafloor spreading[76]. Perhaps the geological age of Aotearoa-New Zealand has not allowed sufficient time for this specialised microorganism to disperse onto the global stage; owing to the multitude of hurdles it must overcome to successfully colonise a new habitat. This scenario, however, does not account for the distribution of *Sulfurihydrogenibium* globally (including Aotearoa-New Zealand, albeit at lower abundances) and suggests other factors such as niche contraction have contributed to this apparent endemism of *Venenivibrio*. Although it is possible ecosystems harbouring *Venenivibrio* populations outside of Aotearoa-New Zealand remain to be sampled, or that some congruence between 16S rRNA genes of Hydrogenothermaceae genera may hide distribution of this taxon when amplicon sequencing is used as a proxy for designating species, our study clearly identifies *Venenivibrio* as a tangible endemic genus within Aotearoa-New Zealand. Given the considerable research on global geothermal springs, particularly from YNP, and the fact we have shown the type strain can grow in global spring water, it seems reasonable to expect reports of *Venenivibrio* if the genus was already established elsewhere.

To conclude, *Venenivibrio* is the dominant bacterial genus found in the microbial communities of Aotearoa-New Zealand geothermal spring water. While the taxon occupies an extreme and isolated niche, local reservoirs are able to facilitate widespread dispersal across the TVZ and tolerance to environmental conditions outside of optimal ranges. Despite globally sourced geothermal springs supporting *Venenivibrio* growth, the taxon has failed to contemporarily distribute beyond Aotearoa-New Zealand, resulting in the establishment of an apparent endemic genus via niche differentiation and allopatric speciation.

Microbial communities are dynamic and environmental fluctuations over time also effect microbial behaviour, ecology and evolution[11,77]. Future studies investigating the longitudinal response of *Venenivibrio* to conditions encountered during dispersal (e.g., environmental stressors and other competitors) would provide a more complete picture of population dynamics. It was previously suggested *V. stagnispumantis* CP.B2[T] was adapted to a moderately acidic environment, hinting that a wider range of pH tolerance precluded the taxon's colonisation of Champagne Pool[28]. Certainly, our results show that while narrow pH and temperature optima increase abundance of the taxon in specific spring environments, tolerance of broader environmental conditions do exist within the genus, signifying the importance of future investigation at species- and strain-level. We must highlight that in only using the 16S rRNA gene of a single strain to search for *Venenivibrio* globally, divergent lineages within the genus might not have been detected. The OTUs and MAGs generated by this study indicate substantial microdiversity within the genus, particularly within Aotearoa-New Zealand, and future research will investigate this concept through the characterisation of multiple *Venenivibrio* isolates and whole genome sequencing. Finally, questions remain over the timeline of diversification within the Hydrogenothermaceae, with preliminary molecular dating suggesting divergence between *Venenivibrio* and *Sulfurihydrogenibium* occurred ~67 mya, and how other family members have successfully managed intercontinental dispersal. Biogeographic patterns of *Sulfurihydrogenibium* have been linked to major geological events in the past[78], so elucidating the role of historic events (i.e., the separation of Zealandia from the Gondwana supercontinent and the Oruanui supereruption of the Taupō volcano)[76,79], tracing gene flow among populations, and determining recombination efficiency will not only further establish the concept of endemism for *Venenivibrio*, but also enhance our understanding of microbial evolutionary processes.

## Methods

### Aquificota and *Venenivibrio* ecology in Aotearoa-New Zealand

The data used to evaluate the distribution of Aquificota and *Venenivibrio* in Aotearoa-New Zealand was generated as part of the 1000 Springs Project[27], a study investigating the biogeography of bacterial and archaeal communities in geothermal springs from the Taupō Volcanic Zone (TVZ). Briefly, microbial communities from the water columns of 925 geothermal springs were determined via 16S rRNA gene amplicon sequencing of the V4 region using the original 515F-806R Earth Microbiome Project primers[51], and the Ion PGM system for next-generation sequencing. Raw sequences were processed using USEARCH v7[80] and QIIME v1.9[81], rarefied to 9500 reads per sample, and taxonomy was assigned to OTUs using the RDP classifier v2.2[82] (with a minimum confidence score of 0.5) and the SILVA SSU v123 database[46]. Forty-six physicochemical parameters were also measured from each spring. These were aluminum (Al), ammonium ($NH_4^+$), arsenic (As), barium (Ba), bicarbonate ($HCO_3^-$), boron (B), bromine (Br), cadmium (Cd), caesium (Cs), calcium (Ca), chloride ($Cl^-$), chromium (Cr), cobalt (Co), conductivity (COND), copper (Cu), dissolved oxygen (dO), ferrous iron ($Fe^{2+}$), hydrogen sulfide ($H_2S$), iron (Fe), lead (Pb), lithium (Li), magnesium (Mg), manganese (Mn), mercury (Hg), molybdate (Mo), nickel (Ni), nitrate ($NO_3^-$), nitrite ($NO_2^-$), oxidation-reduction potential (ORP), pH, phosphate ($PO_4^{3-}$), potassium (K), rubidium (Rb), selenium (Se), silicon (Si), silver (Ag), sodium (Na), strontium (Sr), sulfate ($SO_4^{2-}$), sulfur (S), temperature (TEMP), thallium (Tl), turbidity (TURB), uranium (U), vanadium (V), and zinc (Zn). Conductivity, ORP, dO, turbidity, and pH were determined using a Hanna Instruments multiparameter field meter (Woonsocket, RI, USA), with spring temperature measured by a Fluke 51-II thermocouple (Everett, WA, USA).

Inductively coupled plasma-mass spectrometry (ICP-MS), UV-vis spectrometry, titration, and ion chromatography were used to measure aqueous metals and non-metals. A detailed description of field sampling, sample processing, DNA extraction, DNA sequencing, OTU clustering and classification, and physicochemical analyses can be found in Power et al., 2018[27].

To specifically investigate the ecology of Aquificota and *Venenivibrio* in Aotearoa-New Zealand geothermal springs, statistical analyses were performed using R v4.0.3[83] and the phyloseq package v1.32[84] (unless otherwise stated), with all figures generated using ggplot2 v3.3.2[85]. The 1000 Springs Project dataset was first pruned to only samples containing reads assigned to the Aquificota using the subset_taxa function. Each Aquificota-assigned OTU was agglomerated to its respective genera using the tax_glom function to calculate taxon prevalence over the region, average relative read abundance of each genus across the original 925 springs sampled, and the number of OTUs found per genus. To conservatively analyse Aquificota diversity with respect to environmental variables, exiguous OTUs (not agglomerated to genus) with <20 reads across all samples and samples with <20 Aquificota-assigned reads were filtered from the dataset using the prune_taxa and prune_samples functions, respectively. The number of Aquificota OTUs per spring (i.e., OTU number) and the total abundance of Aquificota-assigned reads per spring (i.e., read abundance) were then calculated. Individual linear regression modelling was applied using Aquificota OTU number and read abundance separately as the response variable, against each physicochemical parameter ($n = 46$) as the predictor variable. This was followed by multiple linear regression using all physicochemical parameters, combined with a backward stepwise Akaike information criterion (AIC), to identify main predictor variables. Correlations of all 46 physicochemical variables against both Aquificota OTU number and read abundance were tested using the dist function to create Euclidean distance matrices, and the cor.test function to calculate Pearson's and Spearman's correlation coefficients. All coefficient *p*-values for both regressions and correlations were corrected using the p.adjust function with the false discovery rate (FDR) Benjamini-Hochberg procedure.

The data was then further pruned to contain *Venenivibrio*-assigned reads only. Again, exiguous OTUs with <20 total reads across all springs were removed, as were springs with <20 reads assigned to *Venenivibrio*. This resulted in 467 springs and 99 OTUs for all subsequent analyses. The number of *Venenivibrio* OTUs per spring (i.e., OTU number) and total read abundance of *Venenivibrio* per spring (i.e., read abundance) were calculated. As detailed for the Aquificota dataset, linear regression modelling and correlations were performed for both *Venenivibrio* diversity metrics against the 46 physicochemical variables measured for each spring.

### Temperature, pH, and distribution of *Venenivibrio* in Aotearoa-New Zealand

To further examine the effect of pH and temperature on these ecosystems, springs were binned by either one pH unit (i.e., springs with pH 4.01–5.00 binned as pH 4–5 etc.) or 10 °C (i.e., springs with temperatures of 60.1–70.0 °C binned as 60–70 °C etc.) increments and plotted against read abundance of *Venenivibrio* in each spring. To calculate a local polynomial regression of read abundance per spring, the loess function with default settings was applied to the data with both pH and temperature as the predictor variables. This model identified maximal abundance of the taxon as ≥45% of the microbial community and so, springs were then partitioned into 'high' (≥45%) and 'low' (<45%) *Venenivibrio* abundance to assess environmental differences between the two subsets. The get_googlemap function from the package ggmap v3.0.0[86] generated maps of the TVZ and individual geothermal fields, with either average relative read abundance in each geothermal field or read abundance in each spring added respectively,

using WGS84 latitude and longitude coordinates. The proportional relative abundance of *Venenivibrio* OTUs per field and spring was also calculated. *Venenivibrio* hotspots were defined as springs with ≥85% of sequencing reads from the microbial community assigned to the genus, and were plotted on an outline of North Island, Aotearoa-New Zealand using the map_data and geom_polygon functions from ggplot2. The vegdist function from vegan v2.5-6[87] was used to generate Bray-Curtis dissimilarities between *Venenivibrio* populations in all springs, which was then coupled with pairwise geographic distance to investigate a distance-decay pattern across the region. Characteristics of each OTU that assigned to the genus *Venenivibrio* were determined, including distribution across the region (i.e., prevalence), average relative read abundance across all 925 springs originally sampled, and associated environmental variables including pH and temperature.

### Genome annotation of *V. stagnispumant*is CP.B2[T]

Extended annotation was performed on the draft genome sequence of CP.B2[T][88] through the Integrated Microbial Genomes annotation pipeline v4.16.4[50] (IMG Taxon ID 2799112217; GenBank Accession: GCA_026108055.1). Hydrogenase annotation was completed using HydDB[89]. Genes of interest from the CP.B2[T] genome were also manually compared to all available Hydrogenothermaceae genomes in IMG using the Function Search tool to identity distinguishing characteristics (via unique genes or metabolic pathways) that might explain the putative exclusivity of *Venenivibrio* to Aotearoa-New Zealand. These nine publicly-available Hydrogenothermaceae genomes were *Hydrogenothermus marinus* VM1[T] (IMG Taxon ID: 2734482257), *Persephonella hydrogeniphila* 29W[T] (2728369220), *P. marina* EX-H1[T] (643692030), *Persephonella* sp. IF05_L8 (2562617012), *Persephonella* sp. KM09_Lau8 (2561511220), *Sulfurihydrogenibium azorense* Az-Fu1[T] (643692050), *S. subterraneum* HGMK1[T] (2558860995), *Sulfurihydrogenibium* sp. YO3AOP1 (642555165), and *S. yellowstonense* SS-5[T] (645058708).

### Growth range reassessment of *V. stagnispumantis* CP.B2[T]

To interrogate the observed widespread distribution of *Venenivibrio* in Aotearoa-New Zealand, specific growth characteristics of the type strain CP.B2[T] were reanalysed including temperature, pH, salinity, and $O_2$ tolerances. Growth media (DSMZ medium 1146) were prepared in 9 mL aliquots (in triplicate) as per Hetzer et al., 2008[28] with the addition of elemental sulfur ($S^0$), inoculated with 1 mL of strain CP.B2[T], and incubated at 70 °C statically for seven days, unless otherwise stated. Both positive and negative controls for each experiment were also prepared accordingly. Phase contrast microscopy was used to determine growth due to the tendency for CP.B2[T] to flocculate. The observed upper and lower growth limits were confirmed by subculturing the culture into the described optimal growth medium and conditions (DSMZ medium 1146; with $S^0$ and 2.5% v/v $O_2$, at 70 °C and pH 5.5) to confirm viability. The growth temperature range for CP.B2[T] was tested between 38.5-79.7 °C using a custom-made temperature gradient oscillator (10 oscillations/min). To test a pH range of 3.0–8.5, DSMZ medium 1146 was prepared without the addition of MES except for where pH 5.5–6.5 was required. To buffer the medium outside this range, either 10 mM sodium citrate/citric acid (pH 3.0–5.0), or 2.38 g/L HEPES (pH 7.0–8.5) buffer were added. The medium pH was adjusted using either 1 M NaOH or 1 M HCl as required. Each enrichment was tested after incubation with both universal and range specific pH paper to ensure consistency of pH throughout the experiment. Salinity growth ranges were tested by the addition of 0.0-10.0% (w/v) NaCl to growth medium. To test the $O_2$ tolerance of CP.B2[T], a headspace of 50:40:10 $N_2$:$H_2$:$CO_2$ was prepared to growth medium, with oxygen then added volumetrically on top of this gas mixture to final concentrations of 1.25, 2.5, 5.0, 7.5, 10, 12.5, 15 and 25% (v/v).

Additionally, geothermal spring water was collected from both Champagne Pool (CP), Waiotapu, Aotearoa-New Zealand, the origin of CP.B2[T], and Obsidian Pool (OP), Yellowstone National Park (YNP), USA,

to test in vitro growth capability of CP.B2$^T$ in a chemically similar, global spring. Both spring water samples were filtered twice, immediately in the field and then in the laboratory, using a Sterivex-GP 0.22 μm PES column filter to ensure sterility, and stored at 4 °C until use. The two spring samples were prepared in triplicate as follows: original spring water with no additions; spring water supplemented with $NH_4Cl$, $KH_2PO_4$, and $S^0$ (as per the concentrations specified in DSMZ medium 1146); and spring water with $S^0$ as the only supplement. In all cases, no trace metals were added to the hybrid spring media. Medium pH did not vary with the addition of amendments. Headspace composition for all experiments was 50:40:7.5:2.5 $N_2$:$H_2$:$CO_2$:$O_2$. An exponential phase CP.B2$^T$ was then inoculated into all test samples and incubated at 70 °C statically for seven days.

## Global search for 16S rRNA genes reported as, or closely related to, *Venenivibrio*

The following nine databases were screened for for both full and partial 16S rRNA gene sequences either reported as, or returning ≥95.0% sequence similarity[90] to *Venenivibrio stagnispumantis* CP.B2$^T$: NCBI's Nucleotide collection (which includes GenBank, EMBL, DDBJ, and RefSeq)[45], the Sequence Read Archive (SRA)[45], SILVA[46], the Ribosomal Database Project (RDP)[47], Greengenes[48], the Integrated Microbial Next Generation Sequencing (IMNGS) platform (which includes amplicon sequencing from SRA, DDBJ, and ENA)[49], JGI's Integrated Microbial Genomes and Microbiomes (IMG/M) system[50], the Earth Microbiome Project (EMP)[51], and the Qiita platform[52]. Criteria for each individual search are reported below.

A pairwise analysis of the full length 16S rRNA gene of *V. stagnispumantis* CP.BP$^T$ (1506 bp; GenBank accession DQ989208.1 or RefSeq accession NR_044029) was undertaken by querying the sequence against the Nucleotide Collection (nr/nt) in NCBI using BLASTN v2.13.0 with the megablast setting (accessed 23/Jul/2021). Returned hits with ≥95.0% sequence similarity and an e-value of zero were investigated.

All samples (both amplicon and metagenomic) from the SRA were screened for reads that assigned to the genus *Venenivibrio* (tax_id=407997) using the Sequence Taxonomic Analysis Tool (STAT) v2.11.2[91] through the Google Cloud BigQuery platform (accessed 06/Dec/2021). As the genome of *V. stagnispumantis* CP.B2$^T$ was not yet in the RefSeq database, only the 16S rRNA gene sequence (RefSeq accession NR_044029) was used by the STAT program to assign taxonomy to k-mers. BLASTN v2.13.0 (megablast setting) was used to confirm any putative similarity of samples to *V. stagnispuma*ntis CP.B2$^T$ as required.

The Browser function in the 16S rRNA gene database SILVA was used to search the SSU r138.1 component (accessed 25/Mar/2022) for entries classified as the genus *Venenivibrio*. The SILVA Incremental Aligner (SINA v1.2.11) was then run with default settings to align any entries found. The search and classify function of SINA was enabled (with default settings) and closest neighbours to *V. stagnispumantis* CP.B2$^T$ (≥95% sequence similarity) in the Ref NR database were identified.

RDP (release 11_6; accessed 25/May/2021) was screened by using Seqmatch to query the 16S rRNA gene sequence of *V. stagnispumantis* CP.B2$^T$. Search settings included type and non-type strains, uncultured and isolate sequences, all sizes (near full length and partial), good quality and KNN matches of 20.

The 16S rRNA gene database Greengenes (v13_8; August 2013) was downloaded through the FTP site (ftp://greengenes.microbio.me/greengenes_release; accessed 02/Jul/2021) and both 97% (97_otu_taxonomy.txt) and 99% (99_otu_taxonomy.txt) OTU representative sets were screened for the presence of Hydrogenothermaceae and *Venenivibrio*. Corresponding sequences were extracted from the accompanying FASTA files, and sequence identity to the 16S rRNA gene of *V. stagnispumantis* CP.B2$^T$ was checked using BLASTN v2.13.0 (megablast setting). Any additional sequences that mapped to these OTUs were

also found in 97_otu_map.txt and 99_otu_map.txt. Finally, Greengenes OTU IDs were cross checked against accession numbers from GenBank to confirm identity (gg_13_5_accessions.txt; https://greengenes.secondgenome.com/; accessed 02/Jul/2021).

The full-length 16S rRNA gene of *V. stagnispumantis* CP.B2$^T$ was queried using the IMNGS platform v1.0 build 2105 (accessed 21/May/2021). This database takes all available raw 16S rRNA gene amplicon samples from the International Nucleotide Sequence Database Collaboration (SRA, DDBJ and ENA), and runs them through a standardised pipeline to produce quality controlled OTUs and corresponding read abundance data. The gene was queried against the full database, using the discrete similarity function at a threshold of 95% sequence similarity (minimum sample size 200 bp). The database was additionally screened for all OTUs assigned to the phylum Aquificae/Aquificota (accessed 25/May/2021). The IMNGS database also included 472 samples from the Tara Oceans project.

All genomes in IMG/M v6.0 (accessed 21/Jul/2021) were searched for Hydrogenothermaceae using the Genomes by Taxonomy function. The 16S rRNA gene of *V. stagnispumantis* CP.B2$^T$ was also searched against all genomes in the database using the BLAST option. Metagenomic samples were screened for Hydrogenothermaceae-assigned bins using the Bins by Taxonomy function under Find Genomes/Metagenome Bins.

The first release of the Earth Microbiome Project (EMP) was searched in multiple ways for the presence of *Venenivibrio*. Firstly, all samples classified using SILVA taxonomy (SSU r123) were downloaded through the EMP FTP site (ftp://ftp.microbio.me/emp/release1; accessed 22/Jun/2021) in the closed-reference dataset emp_cr_silva_16 S_123.release1.biom. This was converted to text and the grep command was used to find OTUs assigned to *Venenivibrio*. The reference OTUs used to assign this taxonomy (silva_123.97_otus_16S.consensus_taxonomy_all_levels.txt) were also checked to confirm presence of *Venenivibrio* in this taxonomy. Secondly, the software redbiom (v0.3.5; accessed 22/Jun/2021) was used to search for *Venenivibrio* in the EMP category (qiita_empo_3) for all contexts (i.e., analyses). As this software is still under active development, the taxon search function is limited to closed reference data using Greengenes taxonomy. The Greengenes database only assigns *Venenivibrio* to the genus level in the 99% OTU representative set (g_Venenivibrio), therefore the corresponding OTU ID from the 97% rep set (OTU ID 32720) was also searched using the feature option. Thirdly, the 16S rRNA gene sequence of *V. stagnispumantis* CP.B2$^T$ was used to search for exact matches in EMP datasets analysed by deblur, again using the feature selection in redbiom. Finally, the EMP subset used to create trading cards (otu_summary.emp_deblur_90 bp.subset_2k.rare_5000.tsv) was checked for *Venenivibrio*. This analysis created 90bp tag sequences using deblur with Greengenes taxonomy.

All publicly available datasets in the Qiita database (https://qiita.ucsd.edu/redbiom/; accessed 05/Jul/2021; version 2021.05 9799e8f) were searched using redbiom by taxon (g_Venenivibrio), feature (OTU IDs 32720 and 1142935), and sequence (16S rRNA gene sequence of *V. stagnispumantis* CP.B2$^T$). The software is still in active development so currently only OTU IDs/taxa names for closed reference data using Greengenes taxonomy or exact sequences for deblur can be searched.

Additionally, a 16S rRNA gene phylogenetic tree was constructed of entries in the SILVA database assigned to the genus *Venenivibrio*. A FASTA file, containing all near-full length trimmed and quality controlled 16 S rRNA gene sequences assigned to the genus *Venenivibrio* in the SILVA database (SSU r138.1; accessed 25-Mar-2022), was imported into the ARB software ecosystem (LTP_09_2021)[92]. Only sequences >1,000 bp in length were included in the analyses to avoid long branch attraction artefacts. In addition, reference sequences (RefSeq) from sister genera, *Sulfurihydrogenibium* and *Persephonella*, plus a selection of Aquificaceae [*Hydrogenobaculum acidophilum* 3H-1$^T$ (D16296), 'Aquifex pyrophilus Kol5a' (M83548), *Hydrogenivirga caldilitoris* IBSK3$^T$

(AB120294), *Hydrogenivirga okinawensis* LS12-2$^T$ (AB235314), *Hydrogenobacter hydrogenophilus* Z-829$^T$ (HE616187), *Hydrogenobacter subterraneus* HGP1$^T$ (AB026268), *Thermothrix azorensis* TM$^T$ (GU233444), *Thermocrinis minervae* CR11$^T$ (LT670846), *Thermocrinis albus* HI 11/12$^T$ (CP001931), *Thermocrinis jamiesonii* GBS1$^T$ (KC526152), and *Thermocrinis ruber* OC 1/4$^T$ (CP007028)] were added for more distantly related *Venenivibrio* sequences. All sequences were aligned to the original, quality controlled, *V. stagnispumantis* CP.B2$^T$ (DQ989208) available in ARB, or as appropriate, *Persephonella* or *Sulfurihydrogenibium* reference sequences. A phylogenetic tree (Figure S6) was constructed using TREE-PUZZLE, a quartet-puzzling maximum-likelihood algorithm with 10,000 puzzling steps (substitution model used: HKY).

Finally, all NCBI databases (including PubMed Central) and Google Scholar were searched for any literature and/or samples containing the word '*Venenivibrio*' (accessed 28/Apr/2022). Any 16S rRNA gene sequences putatively identified as *Venenivibrio* were manually checked for ≥95.0% sequence similarity to *V. stagnispumantis* CP.B2$^T$ using BLASTN v2.13.0 with the megablast setting.

### Screening metagenomes for *Venenivibrio*

We screened all available TVZ geothermal metagenomic samples (n = 16) from 10 previously sampled geothermal springs (i.e., local metagenomes; Supplementary Data 16) for the presence of *Venenivibrio* using the metagenomic classifier Kraken2 v2.0.8[53]. The default Kraken2 database failed to classify Hydrogenothermaceae reads to genus or species level, so a custom database was built using the NCBI Taxonomy database[93] and RefSeq complete bacterial reference library[94] (accessed 4/Sep/2020). The *V. stagnispumantis* genome was not included in RefSeq at the time of analysis, so the type strain genome CP.BP$^T$ was downloaded from IMG and added manually (GOLD Analysis ID Ga0170441; IMG Taxon ID 2724679818). The output from Kraken2 was then searched for '*Venenivibrio*' using grep v2.21. TerrestrialMetagenomeDB[54] release 2.0 (January 2021), a curated and standardised metadata repository with 20,206 terrestrial (non-marine) metagenomes from the databases SRA and MG-RAST, was used to search for *Venenivibrio* signatures in hot springs outside of Aotearoa-New Zealand (i.e., global metagenomes). The following terms were used in the search (accessed 31/Jan/2021); 'hot spring', 'hotspring', 'hydrothermal', and 'geothermal'. Resultant metagenomes were downloaded from either SRA using SRA Tools v2.9.3 or MG-RAST manually. These were classified in the same way as local metagenomes using Kraken2, with the custom database described above.

Four local metagenomes with increased *V. stagnispumantis* read abundance, either by metagenomic or 16S rRNA gene community profiling, were independently aligned to both the type strain CP.B2$^T$ (Ga0170441) and three of the most closely related family members using bowtie2 v2.3.5.1[95] (with default settings). These were identified as *Sulfurihydrogenibium yellowstonense* SS-5$^T$ (RefSeq assembly accession GCF_000173615), *Persephonella hydrogeniphila* 29W$^T$ (GCF_900215515), and *Sulfurihydrogenibium* sp. Y03AOP1 (GCF_000020325). The three local springs with the greatest read abundance of *V. stagnispumantis* CP.B2$^T$ from metagenomic community profiling were Waiotapu No.15 Feature 1, Whakarewarewa Feature 51, and Champagne Pool (Supplementary Data 16), along with a fourth spring called Radiata Pool that had increased read abundance from 16S rRNA gene community analysis. In addition, six global metagenomes returning putative *V. stagnispumantis* signatures were also selected to individually map to the Hydrogenothermaceae genomes. These included two springs from the Kirishima region, Kyushu Island, Japan (SRA Accession DRR163686, DRR163687), Jinata Spring, Shikinejima Island, Japan (SRR7905022), Obsidian Pool, Yellowstone National Park, USA (SRR6049666), Dewar Creek, British Columbia, Canada (SRR5580900), and Great Boiling Spring, Nevada, USA (SRR4027810; Supplementary Data 17). The three samples from Japan had the greatest number of *V. stagnispumantis* reads (26,312-594,730) from community profiling of all global spring metagenomes analysed (n = 188), and while the three other global samples from the USA and Canada had putative traces of *Venenivibrio*, they also represented springs with synonymous physicochemistry to the source location for the type strain CP.B2$^T$ (Champagne Pool, Waiotapu geothermal field, Aotearoa-New Zealand).

Output files were first converted to BAM format using samtools v1.10[96] and then to BED format using the genomecov function in bedtools v2.29.2[97] to import into R. For each metagenomic sample (n = 10), average coverage depth per contig was calculated and plotted against relative genome position to assess evenness across the breadth of the four genomes. The fraction of each contig covered (i.e., coverage breadth) was also determined. De novo assembly was then performed on four local and two global metagenomes using all metagenomic reads sequenced from each sample. This included quality control of sequencing reads and binning into metagenome-assembled genomes (MAGs). The ATLAS v2.6a3 workflow[98] (with default parameters) was implemented for metagenomic samples sequenced by Illumina technology. The sole metagenome sequenced by the Ion PGM system (sample ID P2.0050) was incompatible with the ATLAS workflow, therefore MEGAHIT v1.2.9[99] and MetaBAT 2 v2.10[100] were used for assembly and binning of this sample, respectively. All MAGs were analysed with FastANI v1.32[101] to calculate average nucleotide identity (ANI) to *V. stagnispumantis* CP.B2$^T$ and CheckM v1.0.5[102] was used to determine genome completeness and contamination of matching bins.

A sequencing simulator (InSilicoSeq v1.5.1)[103] created synthetic metagenomes to test the classification and alignment accuracy of *Venenivibrio*, *Sulfurihydrogenibium*, and *Persephonella* sequencing reads. The iss function generated one million reads modelled on Illumina MiSeq sequencing technology from each of the following genomes: *V. stagnispumantis* CP.B2$^T$ (Ga0170441), *Sulfurihydrogenibium* sp. Y03AOP1 (GCA_000020325.1), and *P. hydrogeniphila* 29W$^T$ (GCA_900215515.1), along with one million reads from 10 random genomes in NCBI (accessed 21/Dec/2020). The random selection of NCBI genomes used to generate the synthetic metagenomes were 4.2% *Streptococcus agalactiae* FDAARGOS_670 (CP044090.1), 12.7% *Myroides phaeus* 18QD1AZ29W (CP047050.1), 12.9% *Lactobacillus johnsonii* 3DG (CP047409.1), 12.1% *Enterococcus faecium* V1836 (CP044264.1), 3.2% *Pseudomonas* sp. NP-1 (CP056030.1), 3.0% Candidatus *Sulcia muelleri* PSPU (AP013293.1), 37.8% *Klebsiella variicola* KP2757 (CP060807.1), 3.1% *Serratia surfactantfaciens* YD25 (CP016948.1), 1.4% *Acinetobacter baumannii* VB2107 (CP051474.1), and 9.5% *Stenotrophomonas* sp. WZN-1 (CP021768.1). Seqtk-1.3 (r106)[104] was then used to subsample reads to build eight mock communities. The output FASTQ files were classified using Kraken2 with the custom database and aligned to Hydrogenothermaceae genomes using bowtie2 as previously described.

### Phylogenomics of hydrogenothermaceae

An approximate maximum-likelihood (ML) phylogenomic tree was built using SpeciesTree v2.2.0 in KBase[105] to confirm *Venenivibrio* as a distinct genus within the Hydrogenothermaceae. All publicly available genomes and MAGs from the family were imported from NCBI RefSeq[94], GTDB[106], and IMG/M[50] databases (07/Feb/2022), and the ML tree was generated using FastTree2[107] (which applies the Shimodaira-Hasegawa test with 1000 bootstrap replicates) with multiple sequence alignment of 49 house-keeping COG gene families as defined by SpeciesTree[105]. This also ensured robust classification of all assemblies to aid our search for *Venenivibrio* globally. Four MAGs generated from geothermal springs in Aotearoa-New Zealand (i.e., local) and two MAGs from other locales (i.e., global) were included. Additionally, the GTDB relative evolutionary divergence (RED) score of CP.B2$^T$ was calculated[106] and average nucleotide identity (ANI) scores were generated of the Hydrogenothermaceae in IMG using the Compare Genomes function. Molecular clock analysis using full length 16S rRNA gene sequences of *V. stagnispumantis* CP.B2$^T$ and *S. yellowstonense* SS-5$^T$ was also completed

using TimeTree v5[108,109] to estimate a pairwise divergence time of the most recent common ancestor to the two genera.

## Reporting summary

Further information on research design is available in the Nature Portfolio Reporting Summary linked to this article.

## Data availability

The 925 raw amplicon sequences analysed in this manuscript from the 1,000 Springs Project can be found under the study accession PRJEB24353. Shotgun sequences of the *V. stagnispumantis* CP.B2[T] genome have been deposited at DDBJ/ENA/GenBank under accession JAPEIW010000000, with the assembled genome deposited into Genomes Online Database (GOLD Analysis ID Ga0311387). Associated annotations are available at Integrated Microbial Genomes system (IMG Taxon ID 2799112217) and GenBank (GCA_026108055.1). Microorganisms and the ecosystems with which they interact can be of cultural significance to Māori (the Indigenous People of Aotearoa New Zealand). These microbial ecosystems and the knowledge (*mātauranga*) associated with these systems can be considered *taonga* (a treasure). Additionally, under the Treaty of Waitangi, Māori have the right to retain control over data derived from species over which they hold *kaitiakitanga* (guardianship) and *mana whenua* (customary rights). For this reason, the Aotearoa-New Zealand-associated MAGs (Supplementary Data 19) have been uploaded into the New Zealand-located Aotearoa Genomic Data Repository (https://data.agdr.org.nz/). Requests for access to these data can be initiated under Project ID TAONGA-AGDR00025 (https://doi.org/10.57748/vpk8-zp44) and will be granted on the recommendation of the *iwi* (extended kinship group/tribe) holding *mana whenua* over these data. The Aotearoa Genomic Data Repository (AGDR) has been recently described in a publication[110]. Caveats for data use are defined by *mana whenua*, but broadly, data cannot be on-shared or used for commercial purposes without permission, and there are no authorship requirements. For any additional information, please contact the corresponding authors of this manuscript. The 16S rRNA gene sequence and metagenomic search results (Supplementary Data 13-15), and local and global metagenomes analysed (Supplementary Data 16 and 17) are available as Supplementary Data files.

## Code availability

All custom R scripts used for analyses and figures are available at https://gitlab.com/morganlab/collaboration-1000Springs/1000Springs in the venenivibrio-manuscript directory.

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

## Acknowledgements

The authors wish to acknowledge Ngāti Tahu – Ngāti Whaoa as *mana whenua* (customary rights) of *Venenivibrio stagnispumantis* CP.B2ᵀ and all associated data linked with this strain. In addition, we acknowledge all our landowners and Māori collaborators for access and support of the data collected as part of the 1000 Springs Project. *Mana whenua* is also acknowledged for all organisms and associated data from geothermal features within *rohe* (territories) of *iwi* (Māori tribes) that is presented in this research. Finally, we specifically thank Ngāti Tahu – Ngāti Whaoa for their continued support and enthusiasm for this, and related research projects. The collection of data within the Taupō Volcanic Zone, as part of the 1000 Springs Project (https://1000springs.org.nz/), was funded by a Smart Ideas grant (C05X1203 – Microbial Bioinventory of Geothermal Ecosystems), which was awarded by the Ministry of Business, Innovation, and Employment (MBIE) of the Aotearoa-New Zealand Government to M.B.S. and S.C.C.; J.F.P. was supported by a Te Pū Ao-GNS Science Postgraduate Scholarship under the Geothermal Resources of

New Zealand (GRN) programme, and the Te Whare Wānanga o Waikato-University of Waikato Hilary Jolly Memorial Scholarship for freshwater ecology research in Aotearoa-New Zealand. ALR was funded by the US-National Science Foundation (DEB-1134877), with thanks to Emily St John, John Donanho, Nicole Wagner, and John Kelly for bioinformatics assistance. C.K.L., M.B.S., C.R.C. and H.E.W. were supported by the Aotearoa-New Zealand Marsden Fund (UOW1701 and UOC2201, respectively). D.R.C. and E.S.B. acknowledge support from the US-National Science Foundation (EAR-1820658) and NASA (80NSSC19M0150). The authors also wish to thank David Waite for preliminary phylogenomics of the Hydrogenothermaceae.

## Author contributions

M.B.S., J.F.P., C.R.C., S.C.C. and I.R.M. conceptualised the study. J.F.P., M.B.S., C.R.C., S.C.C. and X.C.M. contributed to the experimental design. J.F.P., D.T.H. and X.C.M. performed bioinformatical and statistical analyses. H.E.W. and M.B.S. performed culture experiments. K.C.L., J.W.M., T.J.E., A.L.R., C.K.L., D.R.C. and E.S.B. provided metagenomic and geothermal spring samples. J.F.P., M.B.S. and C.R.C. wrote the manuscript, with contributions from all co-authors.

## Competing interests

The authors declare no competing interests.

## Additional information

[1]Thermophile Research Unit, Te Aka Mātuatua | School of Science, Te Whare Wānanga o Waikato | University of Waikato, Hamilton 3240, Aotearoa New Zealand. [2]Te Tari Pūhanga Tukanga Matū | Department of Chemical and Process Engineering, Te Whare Wānanga o Waitaha | University of Canterbury, Christchurch 8140, Aotearoa New Zealand. [3]Te Kura Pūtaiao Koiora | School of Biological Sciences, Te Whare Wānanga o Waitaha | University of Canterbury, Christchurch 8140, Aotearoa New Zealand. [4]Te Tari Moromoroiti me te Ārai Mate | Department of Microbiology and Immunology, Te Whare Wānanga o Ōtākou | University of Otago, Dunedin 9054, Aotearoa New Zealand. [5]Te Kura Pūtaiao | School of Science, Te Wānanga Aronui o Tāmaki Makau Rau | Auckland University of Technology, Auckland 1010, Aotearoa New Zealand. [6]School of Geographical & Earth Sciences, University of Glasgow, Glasgow G12 8RZ, UK. [7]Laboratory of Microbiology, Wageningen University & Research, 6708 WE Wageningen, the Netherlands. [8]Department of Biology, Portland State University, Portland, Oregon 97207, USA. [9]Department of Microbiology and Cell Biology, Montana State University, Bozeman, MT 59717, USA. [10]Department of Biostatistics, Harvard T. H. Chan School of Public Health, Boston, MA 02115, USA. ✉e-mail: caryc@waikato.ac.nz; matthew.stott@canterbury.ac.nz

