## [Peer Review File · Nature Communications]

A genus in the bacterial phylum Aquificota appears to be endemic to Aotearoa-New ZealandReviewer #1 (Remarks to the Author):

In Powers et al. the authors further investigate an interesting finding from their previous work in documenting community-wide patterns across 1000 geothermal springs. Notably, they found that an abundant and widespread taxa, *Venenivibrio*, was classified in their sites but absent in global databases. Generally, I find the work thorough as they document environmental distributions across the TVZ, provide genomic analyses to support abiotic factors structuring *Venenivibrio* distributions, validate those genomic and environmental findings with growth data, and search global databases for the presence of *Venenivibrio*, including the assembly of MAGs from metagenomic data. The prevailing conclusion is that *Venenivibrio* is geographically restricted to this regional area due to its specific environmental conditions for growth, while local dispersal facilitates sub-genus diversification evidenced by the ~100 OTUs identified across geothermal springs, also could be referred to as microdiversity and provide more context to these results (see Chase/Martiny. 2018. and/or Larkin/Martiny. 2017).

While of interest, I have a few general comments and questions and additional specific questions that are outlined by section below.

General.

1. Given that prior reports of endemism could be explained by the level of genetic resolution applied (see Cho and Tiedje. AEM. 2000), it's unclear if the totality of the observations reported indicate an endemic genus or just an endemic strain/species. Specifically, it's unclear whether the databases queried have a formalized inclusion of *Venenivibrio* in their databases (e.g., GTDB did not have the type strain included and does not have any results for *Venenivibrio* in their database). Even if we assume EMP or the other databases classifies this genus (did not check all), it's also unclear if the results were related to divergent strains unrelated to the queried type strain (more details below in Methods) due to lack of %ID reported on both OTU and genome mapping from metagenomes. Given the genus could encompass a high degree of genetic differentiation, ranging to 75-80% ANI in some cases, what fraction of these reads might represent divergent strains/species? I understand parsing hundreds of metagenomes is computationally expensive, but kraken is a coarse read mapping tool and it's unclear how these results might be biased by the inclusion of a single strain in the custom reference database.

2. Given the authors classification of 100 OTUs of *Venenivibrio*, why weren't these sequences also queried against some of the 16S reference databases? Also, was the reference strain 16S rRNA gene trimmed to just reflect the same variable region? The 16S rRNA can yield different results based on mapping alone especially if comparing full length to smaller variable regions. I would assume the OTUs would provide a more robust search against these queries since they are the same amplified region. Along these lines, was the EMP database open or closed reference? This could also contribute to the absence of *Venenivibrio* in other datasets. The authors might have accounted for these factors, but it was unclear in the reading whether they were implemented or not.

3. I think it's imperative to distinguish between the multiple processes occurring at multiple spatial scales. There is widespread regional distribution of *Venenivibrio* across hundreds of springs with no evidence for dispersal limitation (distance decay $R^2 = 0$). However, at the same time, there is sub-genera diversification as evidenced by the 100 OTUs compiled and the reconstruction of divergent but related MAGs. This implies there are finer-level processes occurring (maybe biotic interactions of partitioning of pH, temp tolerances?). I think a big opportunity is to explore this part further with the "microdiversity" aspect of the *Venenivibrio* lineages. All the analyses are basically there already and it provides a framework to show local scale processes could promote diversification within this endemic genus across springs. Either way, there are multiple places throughout where these processes need to be clearly distinguished at global, regional, and local scales.

4. The authors argue there is "no indication of genus-level endemism" - just a quick question. How do the authors reconcile that numerous environmental studies publish MAGs with zero relation to reference genomes/MAGs - this indicates that these MAGs represent novel lineages in these environments (even at the order, family, genus levels). Obviously these studies do not put the extensive work like the authors have done here, but trying to apply the same logic.

5. Minor but important. If this genus is endemic to this region and given the GTDB has no genomes in their database, the authors should be required to upload their *Venenivibrio* MAGs to NCBI to increase the genomic catalog of *Venenivibrio* from 1 genome to 5.

Specific Sections.

Abstract

1. lines 36-38. Can the authors reference the prior work here and clarify the number of study sites and the ones *Venenivibrio* is found in. As is, reads a bit confusing. Also, given the emphasis on temp and pH, how are these geothermal springs defined?
2. lines 41-42. What is meant by "enhance habitat isolation"?
3. lines 45-47. The concluding sentence could use some clarification. The geographic isolation is predominant because *Venenivibrio* is found in 100s of springs while at the same time limiting global dispersal to other favorable environments.

Introduction

1. line 50. Geographic isolation does not "enable" - gene flow is reduced allowing for diversification
2. lines 51-54. Is the island biogeography model the most appropriate here as it mainly details species richness. Maybe change to the Wallace Line? Also has geographic relevance here?
3. lines 56-58. I don't agree with the statement "microbial taxa in other non-extreme habitats present no limits to dispersal" - there are numerous population level analyses showing geographic isolation of clades in soils and host-associated habitats.
4. line 72. "island-like habitats" - should also reference the Cadillo-Quiroz paper here, basically tests this idea in archaean populations.
5. line 97. not sure what "exclusivity" refers to? endemism?
6. lines 99-101. I would just highlight that the authors reconstructed MAGs from their sites as well. It provides extra information and validity to the phylogenomic analyses.

Methods - generally well documented

1. Mentioned above but how are the OTUs defined? I find an interesting component is associating the environmental distributions of the OTUs to relate to microdiversity. But it's unclear what genetic level we are working with.
2. Throughout the paper. Why not convert read numbers to relative abundances? It's very difficult to cross-reference the values when it's a lot easier to conceptualize relative abundance numbers. Further, some parts of the manuscript have % abundances. Please make these consistent throughout.
3. General removal of subjective language throughout. For instance, "scrutinize", "easily invalidated", "substantive", and others.
4. lines 184-187. What is meant by "exclusivity" in this case? Could mean pan-genome (although not enough genomes unless you were to include the MAGs as well) or unique genes or metabolic pathways? Please elaborate.
5. lines 233-234. How many metagenomes cleared this search filter? Also, why were marine metagenomes ignored? Plenty of examples of sub-genera differentiation in saline tolerances.
6. lines 239-245. Mentioned above, but what are the mapping thresholds here? Also, were the reads mapped independently to each reference genome or mapped in parallel to the "best" hit? BBMap has a function `bbsplit.sh` to map to multiple references at the same time.
7. lines 252-254. What were the parameters for de novo assemblies?
8. lines 255-257. What methods were used for Illumina platforms? Why the differences?
9. lines 252-257. In addition, were the assemblies performed with just the reads that mapped to the reference genomes or was the entire metagenome assembled?
10. lines 276-277. What was the phylogenetic model used and how many bootstraps were simulated?
11. lines 283-285. The molecular clock analyses are pretty rough, especially using the 16S rRNA gene. What are the calibration points? Why not just use the concatenated alignment from the phylogenomics? What are the error estimates (probably very large)? I would remove altogether.

Results - all figures should be referenced with panels, not just Figure X but as Figure XA

1. lines 290-291. and throughout. A lot of the Results are convoluted and hard to follow at times. For example, rewording here to "340 abundant Aquificota OTUs were detected..." makes it a lot easier to follow the sentence structure.
2. lines 295-298. How is "exiguous" defined?
- 2B. Also the legend in Figure 1B is confusing to me. Is it showing only samples with Aquificota in them? And then colored by diversity? But not abundance of them?

3. Again, I would change to relative abundances from read number. And if I'm reading Figure 2 correctly, then numerous communities were nearly 100% Aquificota, especially in high temp samples?
4. Numbers don't always match up across sections and paragraphs. For example, line 293 and 307 for number of Venenivibrio OTUs. lines 346 and 350 for number of samples.
5. Figure 2. I don't really get the point of this figure altogether. Figure 1 should encompass Aquificota results (panel A could just be a stacked barplot) and move the relevant environmental parameters to a panel C. Then, Figure 3 shows the relevant Venenivibrio results, especially since the relationships were weak in Figure 2.
6. Speaking of all these environmental variables. The authors perform a FDR correction but how are covariates considered? I imagine many of these are highly correlated.
7. Figure 2 is unnecessary and then can remove lines 307-319. I think the only parameter discussed later on is salinity in the discussion?
8. lines 323-326. Another example of unclear wording. As written it seems like only samples where Venenivibrio >50% abundance were considered but I think the authors mean when they are >50% there are certain patterns.
9. lines 327-330. Doesn't this suggest the linear regression models in Figure 2 aren't appropriate? There is a polynomial relationship with peak pH and temp? Can we replot Figures 3A/B to show this instead of these boxplots?
10. Similar to Figure 2, I don't fully understand Figure 4 and what's different than Figure 5? If you move the pH and temp data from Fig4 to Fig5 it shows the same thing?
- 10B. Figure 5B is a bit misleading since it's showing the relative abundance within Venenivibrio not the entire community. But some of the communities are dominated by Aquificota right, like nearly 100%? Could you just scale to entire community relative abundances?
11. lines 338-341. It's unclear whether the distance decay was ran with all Venenivibrio OTUs and their relative abundances or just the 19 most abundant OTUs.
12. line 343. What is the metric for evenness here?
13. lines 368-370. It's unclear whether the analyses are discussing the type strain or the other strains as well. This section switches between them. Also, can there be context given here as well instead of just listing gene pathways? Why are these specific ones of interest?
14. lines 381-383. What are the growth optima? Does this correspond to the results in Figure 3? Even if they can grow at a wider range, they are at way higher abundances within a specific range in the community data.
15. lines 392-397. There lacks details in this section. Without constantly cross-referencing the Methods section, I can't follow very easily.
16. lines 400-402 and 408-410. How were the "community read abundances" or "community composition analyses" calculated, by mapping? If so, not a robust method since it doesn't account for genome sizes. I don't think it's highly relevant to the results but clarification is needed.
17. lines 406-408. Refer to Methods #5. If only 16 extra samples, seems to warrant an investigation?
- 17B. Similarly, why were 4 metagenomes excluded? Went from 4 local and 6 global to 4 and 2.

Discussion.

1. lines 454-456, 472-476, and 510-514. Relevant results that were not highlighted in their sections, or too convoluted to pickup on them. Didn't notice any reference to flagella genomic analyses and the saline concentrations were buried in the list of environmental parameters in Figure 2.
2. lines 458-461. Again see Results #14. They might be capable of growing across a wide range but abundances differ by orders of magnitude within a narrow range.
3. lines 464-467. I think this is an excellent opportunity to discuss sub-genera diversification of "microdiversity" - you observe genus wide distributions, but there could be partitioning among microdiversity lineages.
4. lines 476-479. unsure on the "promoting gene flow" - you do observe sub-genera diversification evidenced by 100 OTUs, this suggests there is depressed gene flow within lineages in this genus
5. lines 488-491. I understand the rationale here but in the same dataset where Venenivibrio is highly abundant there is no detection in sediments, per lines 476-479. I find the lack of presence is likely due to the sampling biases, either from lack of geothermal metagenomes ("only" 188) and/or reference strains to characterize.
6. line 503. remove "disadvantages" and generally revise text here. Highlight environmental

selection and filtering instead.

7. lines 532-535. Could turbidity be related to mixing rather than particles? Could fit in better with the dispersal mechanisms proposed in earlier sections

8. lines 539-543 and 550-553. The lack of SOX pathway could indicate cross-feeding? Or other dependency on community members? Could be an explanation of diversification? Also the 6 new strains and physiological traits seem highly relevant but aren't included in the current paper.

9. lines 585-590. I don't follow here since there are numerous papers showing regional endemism in populations and/or structured by host distributions. I would incorporate the Cho and Tiedje paper here to discuss endemism as a function of genetic resolution.

10. lines 596-599. I'm not convinced by the molecular clock analyses, they are coarse and have massive error bars around the dates. It's unclear how relevant this would be here.

Reviewer #2 (Remarks to the Author):

The manuscript titled "Allopatric speciation in the bacterial phylum Aquificota enables genus-level endemism in Aotearoa-New Zealand," authored by Jean F. Power and colleagues, builds upon their previously published work which focused on characterizing the microbial diversity of approximately 1000 geothermal springs in the Taupō Volcanic Zone of New Zealand using 16S rRNA amplicon sequences. In their earlier study, they observed a high abundance, diversity, and prevalence of Aquificota members. In this current manuscript, the authors aimed to delve deeper into the analysis of Aquificaceae and Hydrogenothermaceae families, specifically exploring *Venenivibrio* OTUs. Their investigation involved a comprehensive statistical analysis correlating these OTUs with physicochemical parameters and geographic distribution.

One of the most notable findings, which forms the foundation of the entire article, is the discovery that *Venenivibrio* genus members were exclusively identified in New Zealand's hot springs and were absent in other databases containing similar environments. This observation suggests a high level of specialization to their biological niche and significant challenges in dispersal via marine or airborne routes. I believe the extensive comparisons made against public databases, particularly considering the inclusion of 1000 geothermal spring samples, strongly support the notion of this genus being endemic. Additionally, studying processes such as speciation, diversification, dispersion, and specialization in microbial communities holds great significance in the field of microbial ecology and evolution.

Overall, the manuscript is well-organized and highly comprehensible, effectively presenting the authors' research findings.

Minor comments:

Line 334-336: Figure 5 does not accurately reflect the relative abundance of *Venenivibrio* OTUs, which ranges from 0.8% to 35% of the total abundance. I recommend that the authors consider modifying the numbers to represent percentages (modify 1000 and 3000 for percentages).

Figure S3: If the circle size represents read abundance, I would like clarification on what the numbers in the legend represent. Do they indicate the number of reads assigned to *Venenivibrio*? The legend currently shows numbers ranging from 1000 to 9000. Please provide additional information to ensure clarity in the figure interpretation.

Figure S4: The authors should specify whether the Bray-Curtis index was calculated using only the relative abundance of *Venenivibrio* OTUs or if it considered the entire microbial population. Specifically, it would be helpful to know if the abundance of *Venenivibrio* OTUs was normalized to 100% for the calculation.

Figure 4: Similar to Figure 5, it would be beneficial to indicate read abundance in percentages. For

instance, the value of 6000 could represent 60% of the total microbial community.

Line 341 – 350: The assessment of relative OTU abundance in individual samples solely based on Figure S5 poses some challenges in terms of clarity and comprehension.

Table S17: The title of the table suggests that it represents relative metagenomic read abundance, while the accompanying table provides information on the number of aligned reads.

Figure 6A: If I understand correctly, the figure represents the fraction of each contig that contains the genome represented by reads in the respective metagenomes. In that case, it may be necessary to modify the y-axis label to avoid misleading the reader into thinking it refers to the fraction of the total genome covered by metagenome reads. Typically, such values are represented based on the entire genome rather than individual contigs. I'm curious, for the *Venenivibrio* contigs that exhibit nearly 100% coverage in the global metagenomes, what types of genes do they encode? Are they essential and conserved genes, such as the 16S gene?

REVIEWER COMMENTS

The authors wish to thank the Reviewers for their insightful comments and consider the manuscript to be in a better final version from the review process. We have provided responses to the comments below and, where appropriate, have highlighted manuscript changes in the rebuttal commentary.

Reviewer #1 (Remarks to the Author):

In Powers et al. the authors further investigate an interesting finding from their previous work in documenting community-wide patterns across 1000 geothermal springs. Notably, they found that an abundant and widespread taxa, *Venenivibrio*, was classified in their sites but absent in global databases. Generally, I find the work thorough as they document environmental distributions across the TVZ, provide genomic analyses to support abiotic factors structuring *Venenivibrio* distributions, validate those genomic and environmental findings with growth data, and search global databases for the presence of *Venenivibrio*, including the assembly of MAGs from metagenomic data. The prevailing conclusion is that *Venenivibrio* is geographically restricted to this regional area due to its specific environmental conditions for growth, while local dispersal facilitates sub-genus diversification evidenced by the ~100 OTUs identified across geothermal springs, also could be referred to as microdiversity and provide more context to these results (see Chase/Martiny. 2018. and/or Larkin/Martiny. 2017).

- The authors thank the Reviewer for their perceptive comments on the manuscript and have added appropriate discussion around microdiversity to the text. More specific details about these additions can be found in the individual author responses below.

While of interest, I have a few general comments and questions and additional specific questions that are outlined by section below.

General.

1. Given that prior reports of endemism could be explained by the level of genetic resolution applied (see Cho and Tiedje. AEM. 2000), it's unclear if the totality of the observations reported indicate an endemic genus or just an endemic strain/species. Specifically, it's unclear whether the databases queried have a formalized inclusion of *Venenivibrio* in their databases (e.g., GTDB did not have the type strain included and does not have any results for *Venenivibrio* in their database). Even if we assume EMP or the other databases classifies this genus (did not check all), it's also unclear if the results were related to divergent strains unrelated to the queried type strain (more details below in Methods) due to lack of %ID reported on both OTU and genome mapping from metagenomes. Given the genus could encompass a high degree of genetic differentiation, ranging to 75-80% ANI in some cases, what fraction of these reads might represent divergent strains/species? I understand parsing hundreds of metagenomes is computationally expensive, but kraken is a coarse read mapping tool and it's unclear how these results might be biased by the inclusion of a single strain in the custom reference database.

- While we did use the 16S rRNA gene of the type strain *Venenivibrio stagnispumantis* CP.B2 to search for any presence of the genus globally, it could be assumed that if other species or strains within the genus were present in global samples, a sequence identity of near 95% to CP.B2^T would most likely have been returned (as per Yarza *et al.*, 2014). In the few cases where trace sequences with >95% sequence identity were found (Supplementary Information Sections 2.3.1 and 2.3.6), these were more closely related to sister genera within the Hydrogenothermaceae or implausible due to extremely low sequence count and inhospitable host environment. It is a possibility that congruence between the 16S rRNA gene sequences of sister genera could be masking identification of *Venenivibrio* in global samples and we highlighted this in the Discussion (original lines 605-609).

The *Venenivibrio* genus could indeed encompass a high degree of genetic differentiation as we see 94.8 to 97.9% ANI of the *Venenivibrio* MAGs to CP.B2^T (original lines 426-428; Table S19),

with 20 of the most abundant OTUs having $\geq 94.1\%$ sequence similarity to the type strain (Table S6). However, it is our opinion that this microdiversity is an indication only of divergent lineages within the genus, as multiple biases exist for the methods used to assemble (e.g., MAGs) and cluster (e.g., OTUs) these sequences which can lead to unreliable biological inferences.

As this manuscript proposes an endemic microbial genus, diversification within the genus was not a main focus. The use of isolates and whole genomes provides a much more thorough route for investigating divergent lineages and this will be explored in our next manuscript, using six new strains of the genus isolated from Aotearoa-New Zealand that are currently being characterised. However, it is our present view that if divergent strains exist outside of Aotearoa-New Zealand, we would have most likely picked up them up in the data analysis conducted as part of this manuscript. Even if divergent lineages are discovered globally in the future, questions will remain on the specific biogeography of these taxa that has led to an abundance of the genus in Aotearoa-New Zealand hot springs against an apparent rarity on a global scale.

We do appreciate and acknowledge the concerns raised by the Reviewer, and so in response, we have added the following text to the future directions paragraph of the conclusion: “***We must highlight that in only using the 16S rRNA gene of a single strain to search for Venenivibrio globally, divergent lineages within the genus might not have been detected. The OTUs and MAGs generated by this study indicate substantial microdiversity within the genus, particularly within Aotearoa-New Zealand, and future research will investigate this concept through the characterisation of multiple Venenivibrio isolates and whole genome sequencing***” (Lines 676-681).

➤ To answer other discrepancies noted in this comment:

- The Reviewer stated, “*it's unclear whether the databases queried have a formalized inclusion of Venenivibrio in their databases (e.g., GTDB did not have the type strain included and does not have any results for Venenivibrio in their database)*”. The Reviewer is correct in that GTDB does not currently have *V. stagnispumantis* CP.B2^T. However, we did not use this database specifically in our global search for *Venenivibrio*. We did add all GTDB genomes that classified as Hydrogenothermaceae to our phylogenomic analysis and confirmed they were not *Venenivibrio*. As outlined in the Supplementary Information (Results Section 2.3), the following eight curated reference databases did include sequences of *Venenivibrio* at the time of searching: EMP, Qiita, SILVA, Greengenes, RDP, NCBI's Nucleotide Collection, IMG/M, and IMG/RS.
- The Reviewer also stated, “*It's also unclear if the results were related to divergent strains unrelated to the queried type strain (more details below in Methods) due to lack of %ID reported on both OTU and genome mapping from metagenomes*”. Percentage identities of both the local and global metagenomes to the reference genomes were reported in Table S18, with excerpts of these results outlined in the Results section (original lines 416-425). We have now also added % ID of each OTU sequence to the 16S rRNA gene of CP.B2^T to Table S6 in the Supplementary Information.
- Reviewer 1 was concerned about the sensitivity of the metagenomic classifier Kraken2. The authors shared these concerns and originally addressed them by aligning ten metagenomes with the greatest number of reads classified as *Venenivibrio* to four Hydrogenothermaceae reference genomes. These alignments provided more rigorous confidence in identifying the presence or absence of *Venenivibrio* in both sets of metagenomes. The percentage coverage of the CP.B2^T genome was extremely low for the global metagenomes (breadth: 0.11 – 10.17%; depth: 0.00-16.21x; Table S18), “***leading us to conclude Venenivibrio, or any divergent strain within this genus, were not present in these metagenomes***”. We have now added this aforementioned sentence to the Results section (Lines 452-456). The opening line to the Results paragraph introducing these alignments has also been altered to confirm testing the classification results. It now reads as follows: “***To test the validity of***

these putative traces from the Kraken2 classification, four local and six global metagenomes with *Venenivibrio*-assigned reads were aligned to both the *V. stagnispumantis* CP.B2^T genome and the most closely related Hydrogenothermaceae (Figure 6A)" (Lines 446-449).

To further assess the sensitivity of Kraken2, we also tested the reported classification of the 16 local and 188 global metagenomes using eight synthetic or mock communities with varying *Venenivibrio*, *Sulfurihydrogenibium*, and *Persephonella* abundances. Detailed methods and results of this can be found in the Supplementary Information (Table S20 and Supplementary Methods Sections 1.5.2 and 2.4) which we cite on lines 470-473 of the manuscript. 99.8% of *Venenivibrio* reads were correctly recovered by Kraken2 from a community with 100% CP.B2^T, with no cross-over to other sister genera observed. Even when mock communities had a mixture of Hydrogenothermaceae genera, *Venenivibrio* reads still did not misclassify. The only notable irregularity was that some *Persephonella* reads classified as *Venenivibrio*.

2. Given the authors classification of 100 OTUs of *Venenivibrio*, why weren't these sequences also queried against some of the 16S reference databases? Also, was the reference strain 16S rRNA gene trimmed to just reflect the same variable region? The 16S rRNA can yield different results based on mapping alone especially if comparing full length to smaller variable regions. I would assume the OTUs would provide a more robust search against these queries since they are the same amplified region. Along these lines, was the EMP database open or closed reference? This could also contribute to the absence of *Venenivibrio* in other datasets. The authors might have accounted for these factors, but it was unclear in the reading whether they were implemented or not.

- We undertook a more comprehensive search for evidence of the genus along the full length 16S rRNA sequence (~1500 bp), not just the shorter, partial sequences (250 bp) of the OTUs. Each database search included both full length and partial length sequences of the 16S rRNA gene, so conceivably any hits >95% of the 250 bp sequence, representative of the OTUs, would have been returned by our search criteria. Also, OTU clustering can induce diversity inflation, so the full-length 16S rRNA gene from a characterised isolate of the genus provided the best confidence in searching both full- and partial-length sequence similarity from global samples. These criteria are outlined in Section 1.4 in the Supplementary Information (Methods), where we have also clarified "**both full and partial 16S rRNA gene sequences**" were searched (Section 1.4, Page 3, Paragraph 1 of Supplementary Information).

The reference strain CP.B2^T was not trimmed for any database queries as both full-length and partial sequences were searched. Additional mapping was performed with the SILVA SSU r138.1 database, where the SILVA Incremental Aligner (SINA v1.2.11) was run on any sequences that initially classified as *Venenivibrio*. Details of this are outlined in the Supplementary Information (Sections 1.4.3 and 2.3.3).

The multiple EMP searches queried either the closed-reference SILVA or closed-reference GreenGenes datasets. The presence of *Venenivibrio* was confirmed in all these reference datasets before searching, as outlined in Sections 1.4.8 and 2.3.8 in the Supplementary Information.

3. I think it's imperative to distinguish between the multiple processes occurring at multiple spatial scales. There is widespread regional distribution of *Venenivibrio* across hundreds of springs with no evidence for dispersal limitation (distance decay $R^2 = 0$). However, at the same time, there is sub-genera diversification as evidenced by the 100 OTUs compiled and the reconstruction of divergent but related MAGs. This implies there are finer-level processes occurring (maybe biotic interactions of partitioning of pH, temp tolerances?). I think a big opportunity is to explore this part further with the "microdiversity" aspect of the *Venenivibrio* lineages. All the analyses are basically there already and it provides a framework to show local scale processes could promote diversification within this endemic genus across

springs. Either way, there are multiple places throughout where these processes need to be clearly distinguished at global, regional, and local scales.

- We agree with the Reviewer that finer level processes could be promoting diversification within the genus across TVZ springs. Identifying these processes would require cultures of different *Venenivibrio* lineages, a sentiment supported by the manuscript by Chase & Martiny, 2018. Here the authors suggest full genome sequences are necessary to elucidate the evolution of microdiversity within a population. To this end, we have already isolated nine divergent strains of the genus from Aotearoa-New Zealand and are in the process of characterising different attributes that could explain the microdiversity indicated by this manuscript. As part of a recently awarded grant (Marsden, Royal Society), we also plan to sequence substantive metagenomes from these springs which would provide more extensive information on both the putative function and biogeographic controls of these taxa.

To highlight the Reviewer's observation about the potential role of microdiversity in defining sub-genus diversification for *Venenivibrio*, we have now acknowledged the concept three times in the text:

1. A complete sentence about microdiversity has been added (lines 501-503), with the reference of Larkin & Martiny, 2017: "***The presence of 111 Venenivibrio OTUs also suggests that there is a high degree of microdiversity within this genus (REF), with diversification most likely initiated by environmental partitioning***".
2. We added the term "***microdiversity***" to lines 577-581, with the reference of Chase & Martiny, 2018.
3. We have also altered the caption of Figure 4 to include the term "***microdiversity***". It now reads: "***Figure 4. Prevalence, read abundance, pH, and temperature of putative Venenivibrio microdiversity via operational taxonomic units (OTUs)***".

Our manuscript focuses on the presence of *Venenivibrio* at two geographic levels: local scale (defined as within Aotearoa-New Zealand), and global scale (defined as outside Aotearoa-New Zealand). This was briefly designated in the Methods and Results section titled "***Screening metagenomes for Venenivibrio***" (Lines 227, 238, 289, 431, 437). As already discussed in this response, we do not expand on finer level scales that may contribute to microdiversity within the genus. Therefore, we suggest that these two definitions are adequate for the purpose of this manuscript.

We do agree with the Reviewer that our scales need to be more clearly distinguished. So, to reduce any ambiguity, we have added an extra definition of the geographic scales used to the end of Introduction. The start of the last paragraph in the Introduction now reads: "***To investigate the possibility of genus-level endemism, we undertook a detailed ecological study of Aquificota and Venenivibrio at a local scale (i.e., within Aotearoa-New Zealand) using both 16S rRNA gene amplicon and shotgun metagenomic data, as well as geothermal spring physicochemical measurements, while also searching for evidence of the Venenivibrio genus at a global scale (i.e., outside Aotearoa-New Zealand) in publicly available DNA sequence databases***" (Lines 96-101).

4. The authors argue there is "no indication of genus-level endemism" - just a quick question. How do the authors reconcile that numerous environmental studies publish MAGs with zero relation to reference genomes/MAGs - this indicates that these MAGs represent novel lineages in these environments (even at the order, family, genus levels). Obviously these studies do not put the extensive work like the authors have done here, but trying to apply the same logic.

- This a good question and one that we grappled with while undertaking this research. We anticipate that other endemic genera or even greater taxonomic rankings are likely to be

discovered, and indeed, we can point to a number of thermophilic isolates and MAGs that we have discovered that have not been reported. However, in these cases, the taxa appear rarely or are rare in NZ, so it is arguable that the reason they aren't detected is that they are also rare globally and just haven't been sampled yet. It would be more difficult to argue endemism in these cases.

What makes our suggestion that *Venenivibrio* genus is indeed endemic, is that it is such a dominant taxon within Aotearoa-New Zealand; > 70% of all hot springs in the Taupō Volcanic Zone support measurable *Venenivibrio* OTUs and ~11.2 % (5.3M of the 47.1M reads) in our 1,000 Springs Project survey classified as *Venenivibrio* (Power *et al.*, 2018). *Venenivibrio* is clearly not rare, and the environmental conditions that support the taxon also appear not to be rare (see our experiment where we successfully grew *Venenivibrio* in hot spring water from Yellowstone National Park (Obsidian Pool) (Lines 403-410, and Supplementary Information Sections 1.3, 2.2 & Table S12).

We do, however, recognise that “no indication of genus-level endemism” is a little strong and so have included the following text to: “**Indeed, endemism has often been reported as a function of genetic resolution (REF) but there has yet to be a description of genus-level endemism of microorganisms in either extreme or non-extreme environments**” (Lines 634-636).

5. Minor but important. If this genus is endemic to this region and given the GTDB has no genomes in their database, the authors should be required to upload their *Venenivibrio* MAGs to NCBI to increase the genomic catalog of *Venenivibrio* from 1 genome to 5.

- We agree with the Reviewer. The genome of *Venenivibrio stagnispumantis* CP.B2^T is in NCBI's RefSeq so it will be added to GTDB in due course. However, while the authors are supporters of open science, we are bound by the agreements with our Māori partners and are strong proponents of indigenous data sovereignty. Our agreements require that we deposit our *Venenivibrio* MAGs in a local data repository. This does not preclude anyone's access to the data, but does require permission to be sought from the indigenous owner for access – we see this as a small additional step. A link to this repository is mentioned in the manuscript under Data Availability (lines 730-734) and in the caption for Table S19.

Specific Sections.

Abstract

1. lines 36-38. Can the authors reference the prior work here and clarify the number of study sites and the ones *Venenivibrio* is found in. As is, reads a bit confusing. Also, given the emphasis on temp and pH, how are these geothermal springs defined?

- As per journal guidelines, references are not permitted in the abstract so we leave that decision to the discretion of the Editor. We have, however, altered the text to reduce the confusion highlighted by the Reviewer and clarified the number of study sites analysed against the exact number where *Venenivibrio* was found. We also included the pH and temperature ranges of the 686 springs where *Venenivibrio* was found to answer the reviewer's second question on spring definition. The lines now read: “**Previous microbial community analysis via 16S rRNA gene sequencing of 925 geothermal springs from the Taupō Volcanic Zone (TVZ), Aotearoa-New Zealand, revealed widespread distribution and abundance of a single bacterial genus across 686 of these ecosystems (pH 1.2-9.6 and 17.4-99.8 °C). Here, we present evidence to suggest that this genus, *Venenivibrio* (phylum Aquificota), is endemic to Aotearoa-New Zealand**” (Lines 34-39).

2. lines 41-42. What is meant by "enhance habitat isolation"?

- The phrase refers to increased isolation of *Venenivibrio* habitats from neighbouring biomes, due to the narrow range of environmental conditions conducive to maximal growth. We have changed the word “*enhances*” to “*increases*” to clarify interpretation (Line 39).

3. lines 45-47. The concluding sentence could use some clarification. The geographic isolation is predominant because *Venenivibrio* is found in 100s of springs while at the same time limiting global dispersal to other favorable environments.

- We have adjusted the concluding sentence accordingly: “***We conclude that geographic isolation, complemented by physicochemical constraints, has resulted in the establishment of an endemic bacterial genus***” (Lines 47-49).

Introduction

1. line 50. Geographic isolation does not “enable” - gene flow is reduced allowing for diversification

- We have changed the sentence to include the Reviewer’s comment: “***A central tenet of biology is that geographic isolation reduces gene flow which can result in the evolution of new species (i.e., allopatric speciation)***” (Lines 52-53). We have also replaced the word “enables” in the manuscript title with “*facilitates*”. It now reads: “***Allopatric speciation in the bacterial phylum Aquificota facilitates genus-level endemism in Aotearoa-New Zealand***”.

2. lines 51-54. Is the island biogeography model the most appropriate here as it mainly details species richness. Maybe change to the Wallace Line? Also has geographic relevance here?

- The authors thank the Reviewer for this suggestion and have added the original reference to the Wallace Line (Wallace, 1863) here. The adjusted sentence reads: “***Historical models have proposed that the physical barrier of the ocean can limit migration to island ecosystems (REFS), provoking the divergence of distinct species that are geographically separated and initiating the formation of endemic taxa***”. (Lines 53-56). We still include a reference to the island biogeography model as we feel this is more specific to the migration processes of island-based ecosystems, like those found in the Aotearoa-New Zealand archipelago.

3. lines 56-58. I don't agree with the statement “microbial taxa in other non-extreme habitats present no limits to dispersal” - there are numerous population level analyses showing geographic isolation of clades in soils and host-associated habitats.

- We agree that the generalisation of this comment was too sweeping and have modified the sentence which now reads: “***Microbial taxa in other non-extreme habitats are presented with fewer limits to dispersal (REFS), reinforcing...***” (Lines 58-59).

4. line 72. “island-like habitats” - should also reference the Cadillo-Quiroz paper here, basically tests this idea in archaean populations.

- We added a sentence about this suggested reference immediately after original line 72: “***Aquificota are also predominantly microaerophilic, further restricting the range of environmental conditions amenable to growth and successful dispersal between ‘island-like’ habitats. Indeed, ecological differentiation has even been suggested to induce sympatric speciation within an archaean population of a single hot spring***” (Lines 72-75).

5. line 97. not sure what “exclusivity” refers to? endemism?

- Yes, we are referring to endemism here so we have clarified our intended interpretation in the text by adding Aotearoa-New Zealand: “***Genome analysis identified distinguishing characteristics that may explain the exclusivity of this taxon to Aotearoa-New Zealand...***” (Lines 101-104).

6. lines 99-101. I would just highlight that the authors reconstructed MAGs from their sites as well. It provides extra information and validity to the phylogenomic analyses.

- We have appended the following text to the aforementioned sentence: "...**which was corroborated by the reconstruction of four *Venenivibrio* MAGs from TVZ springs**" (Lines 104-108).

Methods - generally well documented

1. Mentioned above but how are the OTUs defined? I find an interesting component is associating the environmental distributions of the OTUs to relate to microdiversity. But it's unclear what genetic level we are working with.

- As it is hard to define the exact genotypes the OTUs represent (i.e., species, strain, or sub-strain level), we decided to focus our analyses at the genus level. While investigating microdiversity within this genus, through environmental distributions of individual OTUs, is definitely of interest, we feel that this concept falls outside the scope of the current manuscript. We have isolated nine novel strains of *Venenivibrio* from TVZ springs that are currently being characterised and these isolates, along with the four *Venenivibrio* MAGs generated here, would provide a better avenue for investigating the microdiversity concept proposed by the reviewer in future manuscripts.

2. Throughout the paper. Why not convert read numbers to relative abundances? It's very difficult to cross-reference the values when it's a lot easier to conceptualize relative abundance numbers. Further, some parts of the manuscript have % abundances. Please make these consistent throughout.

- All figures and results have been changed to relative abundance in percent.

3. General removal of subjective language throughout. For instance, "scrutinize", "easily invalidated", "substantive", and others.

- "Scrutinise" has been changed to "examine", "easily" and "substantive" have been removed, and "substantively" has been replaced with "considerably".

4. lines 184-187. What is meant by "exclusivity" in this case? Could mean pan-genome (although not enough genomes unless you were to include the MAGs as well) or unique genes or metabolic pathways? Please elaborate.

- We were referring to the presence or absence of unique genes or metabolic pathways in the CP.B2^T genome, compared to other family members, that might explain why the genus is endemic to Aotearoa-New Zealand. We have adjusted the text to clarify this: "**Genes of interest from the CP.B2^T genome were also manually compared to all available Hydrogenothermaceae genomes in IMG using the Function Search tool to identify distinguishing characteristics (via unique genes or metabolic pathways) that might explain the putative exclusivity of *Venenivibrio* to Aotearoa-New Zealand**" (Lines 189-193).

5. lines 233-234. How many metagenomes cleared this search filter? Also, why were marine metagenomes ignored? Plenty of examples of sub-genera differentiation in saline tolerances.

- The total number of metagenomes screened here ($n=20,206$) was included in the Results section, with 188 of these samples taxonomically classified using Kraken2. We have additionally added this number to the Methods, as per the reviewer's query.

The Terrestrial Metagenome Database that was used for this specific metagenome search did not include marine-associated metagenomes, except for six hydrothermal vent samples. We decided to focus only on surface features for this particular analysis as we proposed hot springs would be the most likely environment to harbour *Venenivibrio* outside of Aotearoa-New Zealand. This also

allowed a tractable number of samples for metagenomic classification where high performance computing (HPC) is required.

However, marine metagenomes *were* included in our screening of SRA using the Sequence Taxonomic Analysis Tool (details found in the Supplementary Methods Section 1.4.2) Additionally, marine amplicon samples and 16S rRNA gene sequences were analysed from each of the other eight databases screened in the section titled “*Global search for 16S rRNA genes reported as, or closely related to, Venerivibrio*”, which included 472 samples from the Tara Oceans Project in the Integrated Microbial NGS platform. More details about these searches can be found in Sections 1.4 and 2.3 in the Supplementary Methods and Supplementary Results, respectively.

6. lines 239-245. Mentioned above, but what are the mapping thresholds here? Also, were the reads mapped independently to each reference genome or mapped in parallel to the "best" hit? BBMap has a function `bbsplit.sh` to map to multiple references at the same time.

- We used the software program bowtie2 to map four local and six global metagenomes to four reference genomes from the Hydrogenothermaceae family. The metagenomic reads from each sample were mapped independently to each of the reference genomes. We have added the word “***independently***” to this line (Line 245) (and “***individually***” to the global metagenomes reference (Line 251)) to address the reviewer’s query.

We are unclear what “mapping thresholds” the Reviewer is referring to here. From each individual alignment of metagenome to reference genome, we calculated both coverage depth and breadth of each genome contig, and presented these results in Figure 6A and Table S18. These alignments were also rigorously tested by mapping mock metagenomic communities (of pre-defined Hydrogenothermaceae abundances) to the reference genomes, as presented in Tables S20 and S21.

7. lines 252-254. What were the parameters for de novo assemblies?

- Default parameters were used for *de novo* assembly which are outlined in the reference provided for the ATLAS workflow (Kieser *et al.*, 2020). ATLAS uses BBTools for quality control, metaSPAdes for assembly, and both MetaBAT2 and maxbin2 for binning. We have added “***with default parameters***” to this sentence (line 261).

8. lines 255-257. What methods were used for Illumina platforms? Why the differences?

- The ATLAS workflow, as per Kieser *et al.*, 2020, was implemented for metagenomic samples sequenced by Illumina technology, with the tools mentioned above. The sole metagenome sequenced by the Ion PGM System, sample ID P2.0050, was incompatible with the ATLAS workflow, therefore MEGAHIT and MetaBAT2 were used for assembly and binning, respectively.

We have adjusted the text to clarify this. It now reads: “***De novo assembly was then performed on four local and two global metagenomes using all metagenomic reads sequenced from each sample. This included quality control of sequencing reads and binning into metagenome-assembled genomes (MAGs). The ATLAS v2.6a3 workflow (REF) (with default parameters) was implemented for metagenomic samples sequenced by Illumina technology. The sole metagenome sequenced by the Ion PGM system (sample ID P2.0050) was incompatible with the ATLAS workflow, therefore MEGAHIT v1.2.9 (REF) and MetaBAT 2 v2.10 (REF) were used for assembly and binning of this sample, respectively***” (Lines 258-265).

9. lines 252-257. In addition, were the assemblies performed with just the reads that mapped to the reference genomes or was the entire metagenome assembled?

- The entire metagenomes were assembled. We have clarified this in the text by adding “**using all metagenomic reads sequenced from each sample**” in brackets.

10. lines 276-277. What was the phylogenetic model used and how many bootstraps were simulated?

- The approximate maximum-likelihood phylogenetic tree was generated using SpeciesTree within the KBase framework, as referenced by Arkin et al (2018). SpeciesTree utilises Fasttree2, which constructs an initial tree with neighbour joining and minimum-evolution nearest-neighbour interchanges (NNIs), followed by minimum-evolution subtree-pruning-regrafting (SPRs) and maximum-likelihood NNIs to further improve the tree. FastTree2 uses the Shimodaira-Hasegawa (SH) test with 1,000 bootstrap replicates to estimate branch confidence. More details about FastTree2 can be found in Price *et al.*, 2010 which we have added to the aforementioned line, along with bootstrapping information. The section now reads: “**An approximate maximum-likelihood (ML) phylogenomic tree was built using SpeciesTree v2.2.0 in KBase (REF) to confirm *Venenivibrio* as a distinct genus within the Hydrogenothermaceae. All publicly available genomes and MAGs from the family were imported from NCBI RefSeq (REF), GTDB (REF), and IMG/M (REF) databases (07/Feb/2022), and the ML tree was generated using FastTree2 (REF) (which applies the Shimodaira-Hasegawa test with 1,000 bootstrap replicates) with multiple sequence alignment of 49 house-keeping COG gene families as defined by SpeciesTree (REF).**” (Lines 281-287).

11. lines 283-285. The molecular clock analyses are pretty rough, especially using the 16S rRNA gene. What are the calibration points? Why not just use the concatenated alignment from the phylogenomics? What are the error estimates (probably very large)? I would remove altogether.

- The molecular clock analysis performed using the TimeTree resource (Kumar *et al.*, 2017) is based on the work of Marin *et al.*, 2017, where a prokaryotic time tree of 11,784 species was constructed with 87 calibration points. These are listed in the Supplementary Table S3 of that publication. We have added Marin *et al.*, 2017, to the sentence to highlight exact intricacies on how the timeline was calculated, as queried by the reviewer.

We agree with the reviewer that the sole use of the 16S rRNA gene for dating diversification is coarse, hence we do not over-emphasise this result in the manuscript. We primarily use the result in the discussion only to hypothesise that the formation of contemporary landmasses via seafloor spreading could have initiated speciation within the Hydrogenothermaceae family, and clearly state more research needs to be conducted to test this rationale.

Results - all figures should be referenced with panels, not just Figure X but as Figure XA

- Done. We have also had to change the arrangement of panels in Figure 3 to correspond with sequential order in the text.

1. lines 290-291. and throughout. A lot of the Results are convoluted and hard to follow at times. For example, rewording here to "340 abundant Aquificota OTUs were detected..." makes it a lot easier to follow the sentence structure.

- We have re-arranged the structure of this sentence as suggested by the Reviewer.

2. lines 295-298. How is "exiguous" defined?

- Exiguous was defined in the Methods section as Aquificota OTUs with <20 reads across all samples and samples with <20 Aquificota-assigned reads. We had added the word “**exiguous**” to this part of the Results to clearly identify this definition.

2B. Also the legend in Figure 1B is confusing to me. Is it showing only samples with Aquificota in them? And then colored by diversity? But not abundance of them?

- Yes, Figure 1B is showing only samples with Aquificota and the symbols are coloured by diversity (Aquificota OTU number). We have adjusted this caption to reduce ambiguity. It now reads: "**All springs that contained Aquificota (n=891) in the TVZ are plotted according to environmental pH and temperature. Diversity in each spring is represented by the number of Aquificota-assigned OTUs in colour (blue [<50 OTUs], green [50-100 OTUs], or red [>100 OTUs]), with data ellipses assuming multivariate t-distribution and a 95 % confidence level**" (Page 32).

3. Again, I would change to relative abundances from read number. And if I'm reading Figure 2 correctly, then numerous communities were nearly 100% Aquificota, especially in high temp samples?

- Read abundances have been changed to relative abundances in percent. The reviewer is correct in identifying numerous communities with nearly 100% Aquificota, as many extreme ecosystems in TVZ geothermal springs are conducive to growth of taxa from this phylum.

4. Numbers don't always match up across sections and paragraphs. For example, line 293 and 307 for number of *Venenivibrio* OTUs. lines 346 and 350 for number of samples.

- Line 293 referred to the unfiltered number of *Venenivibrio* OTUs (n=111) found across 686 geothermal springs in the TVZ. Line 307 referred to the filtered number of *Venenivibrio* OTUs (n=99) found in 467 geothermal springs, on which all subsequent analyses were performed. While we understand that this leads to different numbers across the text, we introduce line 307 as "**a reduced number of geothermal springs...**" (now Line 317) to highlight/remind the reader that filtering of exiguous reads/taxa was performed in the Methods to be conservative with biogeographic analyses. We have added the word "**exiguous**" to the part of the Methods which describes filtering of *Venenivibrio* taxa (Line 154).

Lines 346 and 350 were referring to two different sets of sample numbers: line 346 was the number of springs where OTU_5 was found (n=456) and line 350 was the total number of springs with *Venenivibrio* (n=467) after the removal of exiguous reads/taxa. These are not inconsistencies and reflect the true nature of the data, which is outlined in both the Methods and Results sections.

5. Figure 2. I don't really get the point of this figure altogether. Figure 1 should encompass Aquificota results (panel A could just be a stacked bar plot) and move the relevant environmental parameters to a panel C. Then, Figure 3 shows the relevant *Venenivibrio* results, especially since the relationships were weak in Figure 2.

- Figure 2 displays opposing relationships between the phylum Aquificota and the genus *Venenivibrio* against select physicochemical parameters (i.e., temperature, pH, oxidation-reduction potential, conductivity, and turbidity). The same analyses were performed on both datasets for confident comparison between the results, which highlight inconsistencies between the two taxonomic ranks. The authors consider these discrepancies noteworthy to report, as a weak relationship is not the absence of a result. The specific *Venenivibrio* relationship with pH and temperature is then further scrutinized in the Results section titled "**Temperature, pH, and distribution of *Venenivibrio* in Aotearoa-New Zealand**" and Figure 3. Further, a stronger relationship between *Venenivibrio* with ORP, salinity (via conductivity), and turbidity is observed than with the phylum as a whole in Figure 2, and these findings are also discussed in the text (lines 308-315, 322-329, 550-553, 567, 569-574). Therefore, the authors respectively decline the suggestion by the Reviewer in this instance.

6. Speaking of all these environmental variables. The authors perform a FDR correction but how are covariates considered? I imagine many of these are highly correlated.

- Covariates of the 46 physicochemical parameters measured for each spring were examined in detail in our previous manuscript on whole microbial communities from these geothermal springs

(Power *et al.*, 2018), with a representative chosen from each covariate group based on the greatest mantel statistic with beta diversity and individual variance. This revealed the environmental parameters that had the most correlation with microbial variability in these ecosystems were pH, temperature, ORP, sulphate, nitrate, turbidity, arsenic, ammonium, bicarbonate, hydrogen sulphide, conductivity, lithium, aluminium, silicon, and phosphate. Interestingly, many of these parameters were also associated with both *Venenivibrio* diversity and abundance (e.g., pH, temperature, ORP, turbidity). In summary, the removal of covariates here would not have affected our interpretation of the results.

7. Figure 2 is unnecessary and then can remove lines 307-319. I think the only parameter discussed later on is salinity in the discussion?

- The authors feel that reporting weaker or non-existent relationships between *Venenivibrio* and certain environmental parameters still has merit, in particular as the same analysis was performed against the phylum Aquificota as a whole. This allows a direct comparison between phylum- and genus-level subsets of data. The reviewer is correct in stating that salinity is discussed later in the manuscript, but we also discuss ORP and turbidity in lines 322-329, 567 and 569-574, which demonstrate significant associations with *Venenivibrio* abundance and diversity in the analysis questioned here.

8. lines 323-326. Another example of unclear wording. As written it seems like only samples where *Venenivibrio* >50% abundance were considered but I think the authors mean when they are >50% there are certain patterns.

- Lines 323-326 (now 333-337) refer to samples with >42% abundance of *Venenivibrio* in the microbial communities. This value is now 45% after re-doing the polynomial model with relative abundances. Lines 326-327 (now 336-337), in comparison, refer to samples with <42% abundance of *Venenivibrio*. We have altered the text to remove any ambiguity highlighted by the reviewer and removed the read count so just relative abundance is presented, as is now consistent with the rest of the manuscript. The lines now read: **“From the reduced dataset of 467 springs, microbial communities which contained $\geq 45\%$ of *Venenivibrio*-assigned reads ($n=86$) had a median pH and temperature of 6.0 (IQR 1.3) and 64.8 °C (IQR 15.2), respectively (Figure 3C). In comparison, springs with <45 % of *Venenivibrio* reads ($n=381$) had a median pH and temperature of 6.1 (IQR 4.3) and 61.9 °C (IQR 41.3), respectively”** (Lines 333-337).

9. lines 327-330. Doesn't this suggest the linear regression models in Figure 2 aren't appropriate? There is a polynomial relationship with peak pH and temp? Can we replot Figures 3A/B to show this instead of these boxplots?

- A polynomial relationship between *Venenivibrio* abundance, pH and temperature is already shown in Figure 3C. As the geothermal springs from this dataset span very broad gradients of both pH and temperature, the boxplots displayed in Figures 3A and 3B break these gradients into bins and show more specifically the increments where *Venenivibrio* abundance is maximal. We advocate that the linear regressions models in Figure 2 are still valid as opposing relationships between the selected physicochemistry and the Aquificota phylum are displayed.

10. Similar to Figure 2, I don't fully understand Figure 4 and what's different than Figure 5? If you move the pH and temp data from Fig4 to Fig5 it shows the same thing?

- While the Reviewer is correct in identifying some similarity between the two figures, different information is presented in both. Figure 4 is OTU-orientated and provides characteristics on the microdiversity of *Venenivibrio* populations (via individual OTUs), by displaying the 10 most abundant *Venenivibrio* OTUs (>10% reads per microbial community), the exact number of springs where they are found, and the specific pH and temperature ranges for each abundant OTU. Figure 5 is location-orientated, demonstrating the average total abundance of *Venenivibrio* per

geothermal field (5A) and the proportion or evenness of OTUs spread across each location (5B). Importantly, Figure 5B shows the cosmopolitan distribution of *Venenivibrio* populations across the entire TVZ.

10B. Figure 5B is a bit misleading since it's showing the relative abundance within *Venenivibrio* not the entire community. But some of the communities are dominated by Aquificota right, like nearly 100%? Could you just scale to entire community relative abundances?

- The spread of these *Venenivibrio* OTUs across each geothermal field is not easily interpreted when Figure 5B is scaled to entire community relative abundances. While some springs do harbour nearly 100% Aquificota, many do not and the breakdown of *Venenivibrio* abundance per OTU is not visible when whole communities are displayed in this figure.

To originally address this, Figure S5 was added to the Supplementary Information to show absolute *Venenivibrio* OTU abundances across all 467 microbial communities. When this data is collapsed per geothermal field, assessing these abundances becomes indecipherable. We therefore elect not to change this figure.

11. lines 338-341. It's unclear whether the distance decay was ran with all *Venenivibrio* OTUs and their relative abundances or just the 19 most abundant OTUs.

- All 99 *Venenivibrio*-assigned OTUs (and their relative abundances) were used to assess the distance-decay pattern. We have added this to the sentence, which now reads "***In contrast to a weak distance-decay pattern observed in microbial communities across the TVZ (REF), Venenivibrio populations did not show this trend (n=99 OTUs, R²=0, p=0.007; Figure S4), with 19 of the most abundant OTUs found in all 14 geothermal fields analysed (Figure 5B)***" (Line 350-353).

12. line 343. What is the metric for evenness here?

- No specific diversity metric for evenness (e.g., Shannon diversity index) was calculated here, rather we visually assessed the evenness in the spread of OTUs in Whakaari/White Island from Figure 5A. The authors recognise that this could lead to assumptions from the reader so we have altered the text to reflect this. It now reads: "***In particular, Whakaari/White Island, situated ~48 km offshore from the mainland, showed an increased even spread Venenivibrio OTUs in spring communities when compared to other fields***" (Lines 353-355).

13. lines 368-370. It's unclear whether the analyses are discussing the type strain or the other strains as well. This section switches between them. Also, can there be context given here as well instead of just listing gene pathways? Why are these specific ones of interest?

- We have now split this Genome Annotation section into two paragraphs to improve clarity. The first paragraph solely deals with annotation of the *Venenivibrio* type strain, *V. stagnispumantis* CP.B2^T. The second paragraph highlights notable similarities and differences between CP.B2^T and other Hydrogenothermaceae genomes.

To further reduce any ambiguity, we have added the full name of the type strain when we state that it was included in the comparative analysis between ten Hydrogenothermaceae genomes in the second paragraph. As requested by the reviewer, context to these findings has been added by describing the rTCA cycle fixes carbon dioxide and the paragraph now ends with the line: "***Collectively, these differences suggest a decreased metabolic capability for CP.B2^T compared to other Hydrogenothermaceae isolates***" (Lines 389-391).

14. lines 381-383. What are the growth optima? Does this correspond to the results in Figure 3? Even if they can grow at a wider range, they are at way higher abundances within a specific range in the community data.

- Growth optima are outlined in Table S12 and we have also added these figures to the text. We have altered the sentence to reinforce that the optima do correspond with the data seen in Figure 3. The line now reads: "**However, reassessed growth optima for temperature (70.4 °C), pH (pH 6.0), salinity (0-0.2 % w/v), and O₂ (<1.25-10 % v/v) tolerance did not change substantially (Table S12), corresponding with the preferred conditions for increased abundance of the taxon as demonstrated in Figure 3**" (Lines 400-403).

15. lines 392-397. There lacks details in this section. Without constantly cross-referencing the Methods section, I can't follow very easily.

- We have expanded this section to include more detailed results, with individual database search results outlined in the Supplementary Information. We have also removed the generality of the last line in the main Results section, referring the reader to the Supplementary Information for more details on all results, and have added an individual reference at the end of each appropriate segment.

16. lines 400-402 and 408-410. How were the "community read abundances" or "community composition analyses" calculated, by mapping? If so, not a robust method since it doesn't account for genome sizes. I don't think it's highly relevant to the results but clarification is needed.

- 16 local and 188 global metagenomes were taxonomically classified using the Kraken 2 program against the NCBI RefSeq database, as stated in the Methods section. Additionally, four local and six global metagenomes were then mapped to four Hydrogenothermaceae reference genomes (including the type strain for *Venenivibrio*, *V. stagnispumantis* CP.B2) using bowtie2 to firstly, confirm the Kraken2 results, and secondly, compute coverage breadth and depth of the metagenomic samples across the four reference genomes. We have now clarified that community composition was classified using Kraken2 in the two lines highlighted above in the Results section (now lines 431-434 and 440-441).

17. lines 406-408. Refer to Methods #5. If only 16 extra samples, seems to warrant an investigation?

- As already mentioned, we proposed that surface features would be the most likely environment agreeable to *Venenivibrio* growth, based on growth characteristics for the type strain CP.B2 in the lab, biogeographic analysis from 467 geothermal springs in Aotearoa-New Zealand, and typical hot spring habitats for *Venenivibrio*'s closest related sister genus, *Sulfurihydrogenibium*. Metagenomic classification uses high-performance computing so a tractable number needed to be selected for this analysis. Also, marine metagenomic and amplicon samples were included in two other database searches, namely the Sequence Read Archive and the Integrated Microbial NGS platform, as outlined in the Methods.

17B. Similarly, why were 4 metagenomes excluded? Went from 4 local and 6 global to 4 and 2.

- The authors assume the Reviewer is referring to our generation of metagenome-assembled genomes (MAGs) here. The main purpose of this analysis was to test if near full-length MAGs of *Venenivibrio* could be assembled from the NZ metagenomic samples to further validate taxonomic classification (via Kraken2) and mapping to Hydrogenothermaceae genomes (via bowtie2) of these samples. We also wanted to confirm the absence of *Venenivibrio* in the global metagenomes, so we chose the two global metagenomes with the greatest number of putatively assigned *Venenivibrio* reads from the both the classification and mapping analyses (DRR163686 and DRR163687) to generate MAGs. The closest MAGs generated to *Venenivibrio* from these global metagenomes were 77.8% average nucleotide identity (ANI).

This result satisfied our question that *Venenivibrio* was not present and as the rest of the global metagenomes with putative traces of *Venenivibrio* had even less reads assigned to the genus (<0.03% of the total metagenomic sample) than DRR163686 and DRR163687, along with

<2.18% of the *V. stagnispumantis* CP.B2 genome covered by these reads, we deemed it implausible that *Venenivibrio* MAGs could be generated from the remaining four global metagenomes.

We have added additional text to the sentence introducing this analysis in the Results: **“Additionally, metagenome-assembled genomes (MAGs; Table S19) were created from four local and two global metagenomes with the greatest number of reads putatively assigned (via Kraken2; Tables S16 & S17) and mapped (via bowtie2; Table S18) to *Venenivibrio*”** (now lines 464-466).

Discussion.

1. lines 454-456, 472-476, and 510-514. Relevant results that were not highlighted in their sections, or too convoluted to pickup on them. Didn't notice any reference to flagella genomic analyses and the saline concentrations were buried in the list of environmental parameters in Figure 2.

- Lines 454-456 referred to a result reported in Power et al., 2018, that *Venenivibrio* was identified as the most abundant genus across 925 geothermal springs in the TVZ. We have now referenced this manuscript at the end of this line (now lines 491-493).

Lines 472-476. We thank the reviewer for this observation and we have added the text with **“annotation also confirmed flagellar assembly and a range of chemotaxis genes (Table S11)”** to the Genome Annotation section of the Results (now lines 380-381).

Lines 510-514. Growth of CP.B2^T in NaCl was reported in the Results (lines 379-383 in the original submitted manuscript) and we point the reader to Table S12 for detailed results on this section. In addition to conductivity (as a proxy for salinity) presented in Figure 2, we mentioned Na specifically in lines 324 and 326 in the Results. The Supplementary Tables S7-S10 also contain specific results ascribing to the relationship between salinity (reported as Na) and *Venenivibrio*.

2. lines 458-461. Again see Results #14. They might be capable of growing across a wide range but abundances differ by orders of magnitude within a narrow range.

- We have added the word **“persistence”** to this line as this highlights we are not referring to habitats with maximal abundance of the taxon (line 496).

3. lines 464-467. I think this is an excellent opportunity to discuss sub-genera diversification of "microdiversity" - you observe genus wide distributions, but there could be partitioning among microdiversity lineages.

- We added the following sentence immediately before original lines 464-467, with the reference of Larkin & Martiny, 2017: **“The presence of 111 *Venenivibrio* OTUs also suggests that there is a high degree of microdiversity within this genus (REF), with diversification most likely initiated by environmental partitioning”** (Lines 501-503).

4. lines 476-479. unsure on the "promoting gene flow" - you do observe sub-genera diversification evidenced by 100 OTUs, this suggests there is depressed gene flow within lineages in this genus

- We have changed **“gene flow”** to **“dissemination”** as this is more representative of the intended meaning for the sentence – that the planktonic lifestyle of *Venenivibrio* promotes dispersal of all populations around the TVZ. The author is correct in that we do not yet know the limiting factors of gene flow at a micro-habitat level (Line 518).

5. lines 488-491. I understand the rationale here but in the same dataset where *Venenivibrio* is highly abundant there is no detection in sediments, per lines 476-479. I find the lack of presence is likely due to

the sampling biases, either from lack of geothermal metagenomes ("only" 188) and/or reference strains to characterize.

- Lines 476-479 referred to a different dataset/study on sediments from the same or similar hot springs in the TVZ (our study focused solely on water columns). We also state that *Venenivibrio* was **not reported as an abundant community member** in the sediment study which means detection could be possible in decreased abundance within these sediment samples. Whether sampling bias is the main reason for lack of detection outside of Aotearoa-New Zealand remains to be seen.

6. line 503. remove "disadvantages" and generally revise text here. Highlight environmental selection and filtering instead.

- We have replaced the word "disadvantages" with "**restricts**". This section of the text is discussing the capability of *Venenivibrio* to disperse outside of Aotearoa-New Zealand and we proposed that the isolated geography of the archipelago in the Pacific Ocean is a major contributor to restricting this process. Therefore, we have not revised the rest of the sentence, but we have added the phrase "..., **allowing environmental filtering to limit dispersal globally**", as per the reviewer's suggestion (Lines 541-543).

7. lines 532-535. Could turbidity be related to mixing rather than particles? Could fit in better with the dispersal mechanisms proposed in earlier sections

- The Reviewer raises a valid point but based on observations of these habitats in the field, mixing occurs across all types of geothermal springs in the TVZ (e.g., both acid-sulfate and alkaline-chloride springs). The authors expect turbidity is more likely associated with the pH range (pH 4-6) where maximal abundance of *Venenivibrio* is found – springs within this pH range are typically sourced from both deep hydrothermal fluids and shallower groundwater inputs. This leads to increased contact with neighbouring soils and sediments that result in turbid water columns from precipitation of metal solutes, and the suspension of very fine amorphous ash and clays within these features.

8. lines 539-543 and 550-553. The lack of SOX pathway could indicate cross-feeding? Or other dependency on community members? Could be an explanation of diversification? Also the 6 new strains and physiological traits seem highly relevant but aren't included in the current paper.

- The nine recently isolated strains of *Venenivibrio* are still undergoing characterisation and are not ready to be formally published. They represent the next stage in elucidating the ecological mechanisms of *Venenivibrio* and a subsequent manuscript is being prepared with these findings.

We agree that cross-feeding or dependency with other members of the microbial community could be occurring so we have added the following text in bold: "*Either initial characterisation did not capture the full growth capabilities of the type strain, **some type of cross-feeding or dependency is occurring with other community members**, and/or there is much more diversity and physicochemical preferences at species/strain level represented by the 111 *Venenivibrio*-assigned OTUs found in our analysis*" (Lines 577-581).

9. lines 585-590. I don't follow here since there are numerous papers showing regional endemism in populations and/or structured by host distributions. I would incorporate the Cho and Tiedje paper here to discuss endemism as a function of genetic resolution.

- The manuscript discussed here (Louca, 2022) basically reiterates the same finding stated by the reviewer, that microbial endemism can occur at a very fine genetic resolution. However, analysis of 36,795 whole genomes from 7,000 locations around the world found most microbial species and strains are globally distributed, which greatly increases the novelty of *Venenivibrio* as a putative endemic genus.

We have included Cho & Tiedje, 2000, as suggested by the reviewer: “*Endemism has long been suggested for thermophiles in geothermal springs, where physical constraints and extreme physicochemistry create discrete, isolated microbial islands (REFS). **Indeed, endemism has often been reported as a function of genetic resolution (REF)** but there has yet been a description of genus-level endemism of microorganisms in either extreme or non-extreme environments*” (Lines 634-636).

10. lines 596-599. I'm not convinced by the molecular clock analyses, they are coarse and have massive error bars around the dates. It's unclear how relevant this would be here.

- The Reviewer is correct in stating that molecular clock analyses using the 16S rRNA gene for microbial taxa are extremely coarse, hence we have only introduced this concept in the last paragraph of the Discussion section. It is an approximation only and we do not claim that this analysis is definitive. It initiates an interesting discussion point around the role of geological landmass separation in the diversification of microorganisms especially in NZ, which is extremely relevant to the well-proven macroecology concept that geography can drive the evolution of life on this planet.

Reviewer #2 (Remarks to the Author):

The manuscript titled "Allopatric speciation in the bacterial phylum Aquificota enables genus-level endemism in Aotearoa-New Zealand," authored by Jean F. Power and colleagues, builds upon their previously published work which focused on characterizing the microbial diversity of approximately 1000 geothermal springs in the Taupō Volcanic Zone of New Zealand using 16S rRNA amplicon sequences. In their earlier study, they observed a high abundance, diversity, and prevalence of Aquificota members. In this current manuscript, the authors aimed to delve deeper into the analysis of Aquificaceae and Hydrogenothermaceae families, specifically exploring *Venenivibrio* OTUs. Their investigation involved a comprehensive statistical analysis correlating these OTUs with physicochemical parameters and geographic distribution.

One of the most notable findings, which forms the foundation of the entire article, is the discovery that *Venenivibrio* genus members were exclusively identified in New Zealand's hot springs and were absent in other databases containing similar environments. This observation suggests a high level of specialization to their biological niche and significant challenges in dispersal via marine or airborne routes. I believe the extensive comparisons made against public databases, particularly considering the inclusion of 1000 geothermal spring samples, strongly support the notion of this genus being endemic. Additionally, studying processes such as speciation, diversification, dispersion, and specialization in microbial communities holds great significance in the field of microbial ecology and evolution.

Overall, the manuscript is well-organized and highly comprehensible, effectively presenting the authors' research findings.

- The authors thank the reviewer for their positive comments on the manuscript.

Minor comments:

Line 334-336: Figure 5 does not accurately reflect the relative abundance of *Venenivibrio* OTUs, which ranges from 0.8% to 35% of the total abundance. I recommend that the authors consider modifying the numbers to represent percentages (modify 1000 and 3000 for percentages).

- The average relative abundance of *Venenivibrio* has been changed from read numbers to percentage in Figure 5 (panel A) as requested by both Reviewers.

Figure S3: If the circle size represents read abundance, I would like clarification on what the numbers in the legend represent. Do they indicate the number of reads assigned to *Venenivibrio*? The legend currently shows numbers ranging from 1000 to 9000. Please provide additional information to ensure clarity in the figure interpretation.

- Yes, both the colour and size of circles refers to the number of reads assigned to *Venenivibrio*. We have changed these to percentage to be consistent with the rest of the manuscript. The legend title has also been modified from "*Venenivibrio* abundance (reads)" to "***Venenivibrio* relative abundance (%)**" to aid clarity.

Figure S4: The authors should specify whether the Bray-Curtis index was calculated using only the relative abundance of *Venenivibrio* OTUs or if it considered the entire microbial population. Specifically, it would be helpful to know if the abundance of *Venenivibrio* OTUs was normalized to 100% for the calculation.

- The distance-decay pattern was calculated using only *Venenivibrio* populations, based on the relative abundances of *Venenivibrio*-assigned OTUs. The OTU abundances were not normalised prior to this calculation. We have added "***non-transformed***" before and "***only***" after the phrase "***Venenivibrio* populations**" in the figure caption.

Figure 4: Similar to Figure 5, it would be beneficial to indicate read abundance in percentages. For

instance, the value of 6000 could represent 60% of the total microbial community.

- Done as requested.

Line 341 – 350: The assessment of relative OTU abundance in individual samples solely based on Figure S5 poses some challenges in terms of clarity and comprehension.

- The authors agree that identifying individual samples in Figure S5 where 467 geothermal springs are presented does present some difficulty. OTU_5 (medium blue) at 65 and 77 % relative abundance, as mentioned in line 345, can be observed in the Waiotapu panel of Figure S5. However, OTU_26661 (dark blue), mentioned in original lines 348-349, remains indistinguishable due to the great number of springs sampled from the Rotorua geothermal field. **We have therefore split Rotorua samples into two separate sub-locations (Kuirau Park and Te Whakarewarewa Valley) to aid comprehension.**

In addition, each mention of Figure S5 in the text (original lines 345 and 349) also refers to Figure 4, where the relative abundance of these OTUs is presented more clearly.

Table S17: The title of the table suggests that it represents relative metagenomic read abundance, while the accompanying table provides information on the number of aligned reads.

- The table does represent the relative metagenomic read abundance of *Venenivibrio* found in 21 global metagenomes, presented by both percent and read number of the whole metagenomic community. Six of these samples were then additionally aligned to Hydrogenothermaceae genomes, which are highlighted by an asterisk. We have added the word “**additionally**” to describe these highlighted samples, and “**Table S18**” to identify where the alignment results can be found.

Figure 6A: If I understand correctly, the figure represents the fraction of each contig that contains the genome represented by reads in the respective metagenomes. In that case, it may be necessary to modify the y-axis label to avoid misleading the reader into thinking it refers to the fraction of the total genome covered by metagenome reads. Typically, such values are represented based on the entire genome rather than individual contigs. I'm curious, for the *Venenivibrio* contigs that exhibit nearly 100% coverage in the global metagenomes, what types of genes do they encode? Are they essential and conserved genes, such as the 16S gene?

- We have changed the y-axis label as requested to “**Coverage breadth per genome contig (%)**”. Displaying the spread of contig coverage, rather than averaging to the whole genome, provides the reader with more information in the figure. Additionally, the average coverage of the whole genome or fraction of genome covered by the metagenomic samples is presented in Table S18. To this effect, we have added “**Average coverage breadth and depth across full genomes are presented in Table S18**” to the caption of Figure 6A.

The Reviewer is correct in their assessment that the *Venenivibrio* genes almost fully covered (85-98%) by three global metagenomes are essential and conserved. The same *Venenivibrio* contig covered by samples DRR163686, DRR163687, and SRR7905022 encodes the 5S, 16S, and 23S rRNA genes, along with tRNAs for alanine and isoleucine. The following text has been added to the Results section titled “*Screening metagenomes for Venenivibrio*” to reflect this observation: “**Three of the global metagenomes (sample IDs DRR163686, DRR163687, and SRR7905022) did cover 85-98 % of a single Venenivibrio contig (IMG Scaffold ID 2724812627 and Locus Tag Ga0170441_134; Figure 6),**

which encodes 5S, 16S, and 23S rRNA genes, along with tRNAs for alanine and isoleucine" (Lines 459-462)

Reviewer #1 (Remarks to the Author):

In Powers et al. the authors further investigate an interesting finding from their previous work in documenting community-wide patterns across 1000 geothermal springs. Notably, they found that an abundant and widespread taxa, *Venenivibrio*, was classified in their sites but absent in global databases.

Thank you to the authors for their responses and edits based on my original review. Many of the edits provided clarity on the manuscript. However, I am unsure on how much the revised version addresses my main critiques, mainly addressing processes governing microbial diversification. As the other reviewer also mentioned, the major significance of this paper is the investigation into speciation and diversification, not necessarily the observation of an endemic bacterial lineage. Even in their response, the authors state: "this manuscript proposes an endemic microbial genus, diversification within the genus was not a main focus". As currently presented, the revised version did not expand on the potentially interesting and novel findings that I found compelling in the original draft. Rather, the authors focused on smaller edits to clarify results and streamline the text. Given the SI is comprised of 59 pages (with 21 supplemental tables), it was difficult to track relevant results not presented in the main text, even in places the authors note in their responses are found in the supplement.

Furthermore, as previously mentioned, I'm not convinced that an endemic genus is a novel result warranting publication in Nature Comm. as most assembled MAGs from environmental metagenomes could be inferred to be endemic due to the lack of reference genomes or 16S rRNA sequence in reference databases. This is a major over-simplification, as previously mentioned in my original review, given the authors did substantial analyses to document this genus. Ultimately, I am unconvinced that the present results, without addressing diversification, is expanding on prior knowledge in the field. For example, the community data (from the very impressive sampling of 1000 springs) was presented in a prior report with this subsetting OTUs of interest for detailed figures showing relative abundances. Even the physiological data is constrained to one strain (N=1) questioning the robustness of the physiochemical constraints on growth, especially given the authors refusal to include other strains/genomes in these analyses to save for another publication. Finally, I understand the constraints in making data public due to reasons mentioned in the response, but the classification of an endemic species warrants inclusion in public databases without needing special approval. Without this, it precludes any other group investigating this bacterium from either a genomic or environmental perspective, as it would remain absent in future reference databases.

Reviewer #2 (Remarks to the Author):

The authors have revised and modified the manuscript according to the reviewer comments and all my concerns have been addressed.

REVIEWERS' COMMENTS

Reviewer #1 (Remarks to the Author):

In Powers et al. the authors further investigate an interesting finding from their previous work in documenting community-wide patterns across 1000 geothermal springs. Notably, they found that an abundant and widespread taxa, *Venenivibrio*, was classified in their sites but absent in global databases.

Thank you to the authors for their responses and edits based on my original review. Many of the edits provided clarity on the manuscript. However, I am unsure on how much the revised version addresses my main critiques, mainly addressing processes governing microbial diversification. As the other reviewer also mentioned, the major significance of this paper is the investigation into speciation and diversification, not necessarily the observation of an endemic bacterial lineage. Even in their response, the authors state: "this manuscript proposes an endemic microbial genus, diversification within the genus was not a main focus". As currently presented, the revised version did not expand on the potentially interesting and novel findings that I found compelling in the original draft. Rather, the authors focused on smaller edits to clarify results and streamline the text. Given the SI is comprised of 59 pages (with 21 supplemental tables), it was difficult to track relevant results not presented in the main text, even in places the authors note in their responses are found in the supplement.

The authors thank Reviewer 1 for providing further critique to the manuscript. As the study is the first to suggest an endemic microbial genus, we focused on diversification between sister genera within the family Hydrogenothermaceae, not on diversification within the genus *Venenivibrio* itself. We highlighted genomic differentiation between these sister genera (*via* comparative genomics and phylogenomics), while additionally investigating the geographic and environmental aspects that could have induced and promoted diversification within the family. These aspects are thoroughly discussed in the text. The authors still advocate diversification or "microdiversity" within the genus *Venenivibrio* is outside the scope of this current manuscript and will be dealt with in subsequent research. We note here, and below, that we have removed the text about other *Venenivibrio* isolates from the manuscript, which will be subject of future studies related to microdiversity in this genus.

We agree with the Reviewer that the substantial number of supplementary materials provide some difficulty in tracking relevant results. Adding even more results through analysis of diversification within the genus would further complicate an already data-rich manuscript. After adding supplementary methods to the main methods, and changing supplementary tables to supplementary data files as per editorial changes, the supplementary file is now reduced to 21 pages and is much easier to follow.

Furthermore, as previously mentioned, I'm not convinced that an endemic genus is a novel result warranting publication in Nature Comm. as most assembled MAGs from environmental metagenomes could be inferred to be endemic due to the lack of reference genomes or 16S rRNA sequence in reference databases. This is a major over-simplification, as previously mentioned in my original review, given the authors did substantial analyses to document this genus. Ultimately, I am unconvinced that the present results, without addressing diversification, is expanding on prior knowledge in the field. For example, the

community data (from the very impressive sampling of 1000 springs) was presented in a prior report with this subsetting OTUs of interest for detailed figures showing relative abundances. Even the physiological data is constrained to one strain (N=1) questioning the robustness of the physiochemical constraints on growth, especially given the authors refusal to include other strains/genomes in these analyses to save for another publication. Finally, I understand the constraints in making data public due to reasons mentioned in the response, but the classification of an endemic species warrants inclusion in public databases without needing special approval. Without this, it precludes any other group investigating this bacterium from either a genomic or environmental perspective, as it would remain absent in future reference databases.

We acknowledge the Reviewer's point regarding any MAGs being considered endemic due to lack of reference genomes/16S rRNA gene sequences in databases, but disagree that this opinion applies to the narrative of this manuscript. While we acknowledge that novel MAGs could be new genera and also be endemic to the country of origin, these MAGs are not characterised and (currently) do not have the same volume of evidence that supports endemicity that we have provided in this manuscript. Most importantly, the majority of MAGs found to be novel are due to rarity, whereas we clearly demonstrate that *Venenivibrio* is both widespread and abundant within Aotearoa-New Zealand. In our opinion, this makes the study suitable for publication in Nature Communications.

The Reviewer is correct that the data used in this manuscript was collected during our prior study (Power et al., 2018), where analyses focused entirely at the community level. This current study has provided additional, stand-alone analyses on phylum- and genus-specific data, with results specifically relevant to our interpretation of genus-level endemism occurring within the Hydrogenothermaceae family.

Additionally, the genome of the type strain for *Venenivibrio*, *V. stagnispumantis* CP.B2^T, is currently available in NCBI and IMG/G, which allows the genus to be investigated by other groups and thus will not preclude its use in future reference databases. The four local MAGs generated for this study are deposited in the Aotearoa Genomic Data Repository for reasons stated in the Data Availability statement; we defer to the Editor/Nature Communications with respect to the suitability for this arrangement.

Finally, we acknowledge that reporting that we have isolated novel *Venenivibrio* strains creates an expectation that characterisation data is made public as part of this manuscript. In response, we have removed all mentions of these additional isolates from the manuscript text. We have also modified the title to align with editorial suggestions.

Reviewer #2 (Remarks to the Author):

The authors have revised and modified the manuscript according to the reviewer comments and all my concerns have been addressed.

The authors thank Reviewer 2 for their positive and concise assessment.